# SAN-Diff: Structure-aware noise for super-resolution diffusion model

## Abstract

Recent advances in diffusion models, like Stable Diffusion, have been shown to significantly improve performance in image super-resolution (SR) tasks. However, existing diffusion techniques often sample noise from just one distribution, which limits their effectiveness when dealing with complex scenes or intricate textures in different semantic areas. With the advent of the segment anything model (SAM), it has become possible to create highly detailed region masks that can improve the recovery of fine details in diffusion SR models. Despite this, incorporating SAM directly into SR models significantly increases computational demands. In this paper, we propose the SAN-Diff model, which can utilize the fine-grained structure information from SAM in the process of sampling noise to improve the image quality without additional computational cost during inference. In the process of training, we encode structural position information into the segmentation mask from SAM. Then the encoded mask is integrated into the forward diffusion process by modulating it to the sampled noise. This adjustment allows us to independently adapt the noise mean within each corresponding segmentation area. The diffusion model is trained to estimate this modulated noise. Crucially, our proposed framework does NOT change the reverse diffusion process and does NOT require SAM at inference. Experimental results demonstrate the effectiveness of our proposed method, which exhibits the fewest artifacts compared to other generated models, and surpassing existing diffusion-based methods by 0.74 dB at the maximum in terms of PSNR on DIV2K dataset.

## 1 Introduction

Single-image super-resolution (SR) has remained a longstanding research focus in computer vision, aiming to restore a high-resolution (HR) image based on a low-resolution (LR) reference image. The applications of SR span various domains, including mobile phone photography (Ignatov et al., 2022), medical imaging (Huang et al., 2017; Isaac & Kulkarni, 2015), and remote sensing (Wang et al., 2022a; Haut et al., 2018). Considering the inherently ill-posed nature of the SR problem, deep learning models (Dong et al., 2014; Kim et al., 2016; Chen et al., 2021) have been employed. These models leverage deep neural networks to learn informative hierarchical representations, allowing them to effectively approximate HR images.

Conventional deep learning-based SR models typically process an LR image progressively through CNN blocks (Zhang et al., 2018a) or transformer blocks (Liang et al., 2021; Chen et al., 2021; 2023). The final output is then compared with the corresponding HR image using distance measurement (Dong et al., 2014; Zhang et al., 2018a) or adversarial loss (Ledig et al., 2017; Wang et al., 2018b). Despite the significant progress achieved by these methods, there remains a challenge in generating satisfactory textures (Li et al., 2023). The introduction of diffusion models (Ho et al., 2020a; Rombach et al., 2022a) marked a new paradigm for image generation, exhibiting remarkable performance. Motivated by this success, several methods have incorporated diffusion models into the image SR task (Saharia et al., 2022b; Li et al., 2022; Shang et al., 2023; Xia et al., 2023). Saharia *et al.* (Saharia et al., 2022b) introduced diffusion models to predict residuals, enhancing convergence speed. Building upon this framework, Li *et al.* (Li et al., 2022) further integrated a frequency domain-based loss function to improve the prediction of high-frequency details.

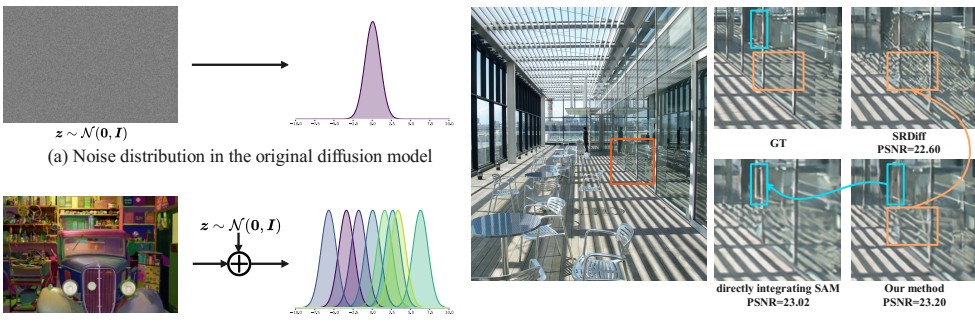

(a) Noise distribution in the original diffusion model

$z \sim \mathcal{N}(\mathbf{0}, \boldsymbol{I})$

(b) Noise distribution in our method

(A) Comparison of noise distribution in the forward diffusion process.

(B) Visualization of restored images generated by different methods.

Figure 1: (A) is comparison of noise distribution in the forward diffusion process between existing diffusion-based image SR methods and our SAN-Diff. Our approach enhances the restoration of different image areas by modulating the corresponding noise with guidance from segmentation masks generated by SAM. (B) is Visualization of restored images generated by different methods. Our method can achieve similar reconstruction performance to directly integrating SAM into diffusion model.

In comparison with traditional CNN-based methods, diffusion-based image SR has shown significant performance improvements in texture-level prediction. However, existing approaches in this domain often employ independent and identically distributed noise during the diffusing process, ignoring the fact that different local areas of an image may exhibit distinct data distributions. This oversight can lead to inferior structure-level restoration and chaotic texture distribution in generated images due to confusion of information across different regions. In the visualization of SR images, this manifests as distorted structures and bothersome artifacts.

Recently, the segment anything model (SAM) has emerged as a novel approach capable of extracting exceptionally detailed segmentation masks from given images (Kirillov et al., 2023). For instance, SAM can discern between a feather and beak of a bird in a photograph, assigning them to distinct areas in the mask, which provides a sufficiently fine-grained representation of the original image at the structural level. This structure-level ability is exactly what diffusion model lacks. But directly integrating SAM into diffusion model may result in significant computational costs at inference stage. Motivated by these problems, we are intrigued by the question: *Can we introduce structure-level ability to distinguish different regions in the diffusion model, ensuring the generation of correct texture distribution and structure in each region without incurring additional inference time?*

In this paper, we verified the feasibility of controlling the generated images by modulating the distribution of noise during training stage, and the theory is illustrated in Figure 1(A). Based on this theory, we proposed the structure-modulated diffusion framework named SAN-Diff for image SR task. This framework utilizes the fine-grained structure segmentation ability to guide image restoration. By enabling the denoise model (U-Net) to approximate the SAM ability, it can modulate the structure information into the noise during the diffusion process.

The training and inference process are illustrated in Figure 3(b). Our method does not change the inference process, and the training process is as follows: (1)For each HR image in the training set, SAM is employed to generate a fine-grained segmentation masks. (2) Subsequently, the Structural Position Encoding (SPE) module is introduced to incorporate masks by position information and generate SPE mask. (3) Finally, the SPE mask is utilized to modulate the mean of the diffusing noise in each fine-grained area separately, thereby enhancing accuracy of structure and texture distribution during the forward diffusion process.

To achieve the goal of reducing the cost of training and inference, our method have with the following advantages:

- During the training, our method *have negligible extra training cost*. We use SAM to pre-generated mask of training samples, and reused them in all epochs. And the cost of modulate noise process is negligible.

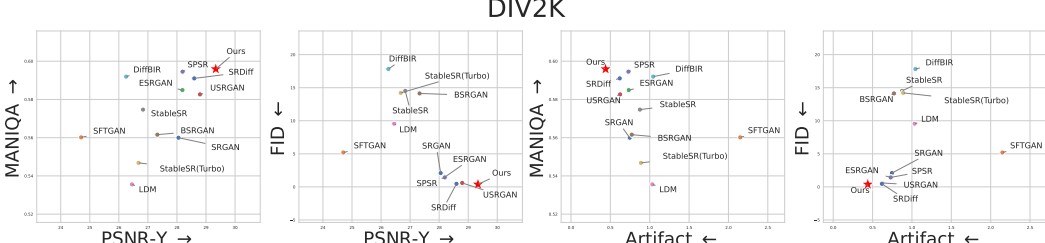

Figure 2: We compared the metrics MANIQA, FID, PSNR, and Artifact(5.3) on the DIV2K dataset. In this context, higher values of MANIQA and PSNR are better, while lower values of FID and Artifact are preferred. The red arrow indicates the direction of the best performance based on the combined horizontal and vertical metrics.

- During the inference, our method *have no additional inference cost*. The diffusion model has already acquired structure-level knowledge during training, it can restore SR images without requiring access to the oracle SAM.

We conduct extensive experiments on several commonly used image SR benchmarks, and our method showcases superior performance over existing diffusion-based methods. Furthermore, our method has the fewest artifacts in generated models such as GAN and diffusion models. Our model achieved a balanced advantage across various metric combinations, as shown in Figure 2.

## 2 RELATED WORKS

### 2.1 DISTANCE-BASED SUPER-RESOLUTION

Neural network-based methods have become the dominant approach in image super-resolution (SR). The introduction of convolutional neural networks (CNN) to the image SR task, as exemplified by SRCNN (Dong et al., 2015), marked a significant breakthrough, showcasing superior performance over conventional methods. Subsequently, numerous CNN-based networks has been proposed to further enhance the reconstruction quality. This is achieved through the design of new residual blocks (Ledig et al., 2017) and dense blocks (Wang et al., 2018b; Zhang et al., 2018b). Moreover, the incorporation of attention mechanisms in several studies (Dai et al., 2019; Mei et al., 2021) has led to notable performance improvements.

Recently, the Transformer architecture (Vaswani et al., 2017) has achieved significant success in the computer vision field. Leveraging its impressive performance, Transformer has been introduced for low-level vision tasks (Tu et al., 2022; Wang et al., 2022b; Zamir et al., 2022). In particular, IPT (Chen et al., 2021) develops a Vision Transformer (ViT)-style network and introduces multi-task pre-training for image processing. SwinIR (Liang et al., 2021) proposes an image restoration Transformer based on the architecture introduced in (Liu et al., 2021). VRT (Liang et al., 2022b) introduces Transformer-based networks to video restoration. EDT (Li et al., 2021) validates the effectiveness of the self-attention mechanism and a multi-related-task pre-training strategy. These Transformer-based approaches consistently push the boundaries of the image SR task.

### 2.2 GENERATIVE SUPER-RESOLUTION

To enhance the perceptual quality of SR results, Generative Adversarial Network (GAN)-based methods have been proposed, introducing adversarial learning to the SR task. SRGAN (Ledig et al., 2017) introduces an SRResNet generator and employs perceptual loss (Johnson et al., 2016) to train the network. ESRGAN (Wang et al., 2018b) further enhances visual quality by adopting a residual-in-residual dense block as the backbone for generator.

In recent times, diffusion models (Ho et al., 2020a) have emerged as influential in the field of image SR. SR3 (Saharia et al., 2022b) and SRdiff (Li et al., 2022) have successfully integrated diffusion models into image SR, surpassing the performance of GAN-based methods. Additionally, Palette (Saharia et al., 2022a) draws inspiration from conditional generation models (Mirza & Osindero, 2014) and introduces a conditional diffusion model for image restoration. Despite their success, generated models often suffer from severe perceptually unpleasant artifacts. SPSR (Ma

et al., 2020) addresses the issue of structural distortion by introducing a gradient guidance branch. LDL (Liang et al., 2022a) models the probability of each pixel being an artifact and introduces an additional loss during training to inhibit artifacts.

## 2.3 SEMANTIC GUIDED SUPER-RESOLUTION

As image SR is a low-level vision task with a pixel-level optimization objective, SR models inherently lack the ability to distinguish between different semantic structures. To address this limitation, some works introduce segmentation masks generated by semantic segmentation models as conditional inputs for generated models. For instance, (Gatys et al., 2017) utilizes semantic maps to control perceptual factors in neural style transfer, while (Ren et al., 2017) employs semantic segmentation for video deblurring. SFTGAN (Wang et al., 2018a) demonstrates the possibility of recovering textures faithful to semantic classes. SSG-RWSR (Aakerberg et al., 2022) utilizes an auxiliary semantic segmentation network to guide the super-resolution learning process.

Image segmentation tasks have undergone significant evolution in recent years, wherein the most recent development is the SAM (Kirillov et al., 2023), showcasing superior improvements in segmentation capability and granularity. The powerful segmentation ability of SAM has opened up new ideas and tools for addressing challenges in various domains. For instance, (Xiao et al., 2023) leverages semantic priors generated by SAM to enhance the performance of image restoration models. Similarly, (Lu et al., 2023) improves both alignment and fusion procedures by incorporating semantic information from SAM. However, these approaches necessitate segmentation models to provide semantic information during inference, resulting in much higher latency. In contrast, our method endows SR models with the ability to distinguish different semantic distributions in images without incurring additional costs at inference.

## 3 PRELIMINARY

### 3.1 DIFFUSION MODEL

The diffusion model is an emerging generative model that has demonstrated competitive performance in various computer vision fields (Ho et al., 2020a; Rombach et al., 2022a). The basic idea of diffusion model is to learn the reverse of a forward diffusion process. Sampling in the original distribution can then be achieved by putting a data point from a simpler distribution through the reverse diffusion process. Typically, the forward diffusion process is realized by adding standard Gaussian noise to a data sample $\boldsymbol{x}_0 \in \mathbb{R}^{c \times h \times w}$ from the original data distribution step by step:

$$q(\boldsymbol{x}_t|\boldsymbol{x}_{t-1}) = \mathcal{N}(\boldsymbol{x}_t; \sqrt{1-\beta_t}\boldsymbol{x}_{t-1}, \beta_t\mathbf{I}), \tag{1}$$

where $\boldsymbol{x}_t$ represents the latent variable at diffusion step $t$. The hyperparameters $\beta_1, \ldots, \beta_T \in (0, 1)$ determine the scale of added noise for $T$ steps. With a proper configuration of $\beta_t$ and a sufficiently large number of diffusing steps $T$, a data sample from the original distribution transforms into a noise variable following the standard Gaussian distribution. During training, a model is trained to learn the reverse diffusion process, *i.e.*, predicting $\boldsymbol{x}_{t-1}$ given $\boldsymbol{x}_t$. At inference time, new samples are generated by using the trained model to transform a data point sampled from the Gaussian distribution back into the original distribution.

As illustrated in Equation 1, identical Gaussian noise is added to each pixel of the sample during the forward diffusion process, indicating that all spatial positions are treated equally. Existing approaches (Saharia et al., 2022b; Li et al., 2022; Shang et al., 2023; Xia et al., 2023) introduce the diffusion model into the image SR task following this default setting of noise. However, image SR is a low-level vision task aiming at learning a mapping from the LR space to the HR space. This implies that data distributions in corresponding areas of an LR image and an HR image are highly correlated, while other areas are nearly independent of each other. The adoption of identical noise in diffusion-based SR overlooks this local correlation property and may result in an inferior restoration of structural details due to the confusion of information across different areas in an image. Therefore, injecting spatial priors into diffusion models to help them learn local projections is a promising approach to improve diffusion-based image SR.

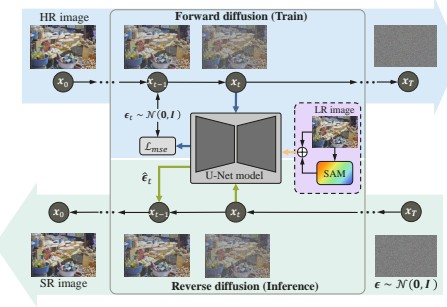 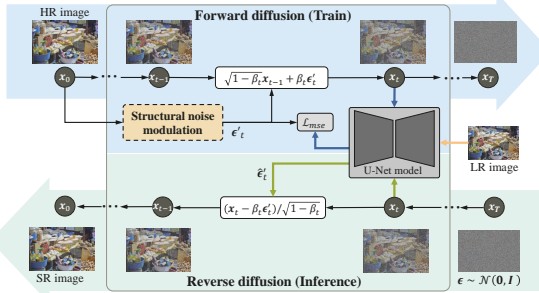

| Parameters: 644M, PSNR: 29.41 | Parameters: 12M, PSNR: 29.34 |
|:---:|:---:|
| (a) Directly integrating SAM | (b) Our propose SAN-Diff |

Figure 3: Comparison between (a) directly integrating SAM into the diffusion model and (b) our proposed SAN-Diff reveals distinct approaches, and the PSNR evaluate on DIV2K dateset. In (a), mask information predicted by SAM is utilized during both the training and inference stages. In contrast, (b) only employs modulated noise generated by the structural noise modulation model during training. The details of structural noise modulation can by found in Figure 4(a), and our method achieves comparable reconstruction performance to (b) as demonstrated in Figure 1(B).

## 3.2 SEGMENT ANYTHING MODEL

Segment Anything Model (SAM) is proposed as a foundational model for segmentation tasks, comprising a prompt encoder, an image encoder, and a lightweight mask decoder. The mask decoder generates a segmentation mask by incorporating both the encoded prompt and image as input.

In comparison to conventional cluster-based models and image segmentation models, SAM is preferable for generating segmentation masks in image SR tasks. Cluster-based models lack the ability to extract high-level information from images, resulting in the generation of low-quality masks. Deep-learning image segmentation models, while capable of differentiating between different objects, produce coarse masks that struggle to segment areas within an object. In contrast, SAM excels in generating extraordinarily fine-grained segmentation masks for given images, owing to its advanced model architecture and high-quality training data. It can generate mask for each different texture region. This ability to distinguish different texture distribution is we aspire to incorporate into diffusion model.

Table 1: Comparison of the effectiveness and efficiency of various diffusion-based image super-resolution methods.

|  | SRDiff | SAM+SRDiff | SAN-Diff |
|:---:|:---:|:---:|:---:|
| Parameter | 12M | 632M+12M | 12M |
| Train time | 10h16min/100k step | 48h52min/100k step | 10h21min/100k step |
| Inference time | 37.64s/per img | 65.72s/per img | 37.62s/per img |
| PSNR | 28.6 | 29.41 | 29.34 |
| FID | 0.4649 | 0.3938 | 0.3809 |

## 3.3 DIRECTLY INTEGRATING SAM INTO DIFFUSION MODEL

To validate the enhancing effect of structure level information on the diffusion process, we devised a straightforward diffusion model (SAM+SRDiff) to utilize the mask information predicted by SAM. Specifically, we concatenated the LR image with the embedding mask information to guide the denoising model in predicting noise. The model structure is detailed in Figure 3(a). Results indicate that the images generated by this simple model exhibit more accurate texture and fewer artifacts.

However, this approach introduces additional inference time as SAM predicts the mask, as shown in Table 1. Can we enable the diffusion model to learn the capability of distinguishing different texture distributions without relying on an auxiliary model? Furthermore, is it possible to train the denoising model to acquire this capability?

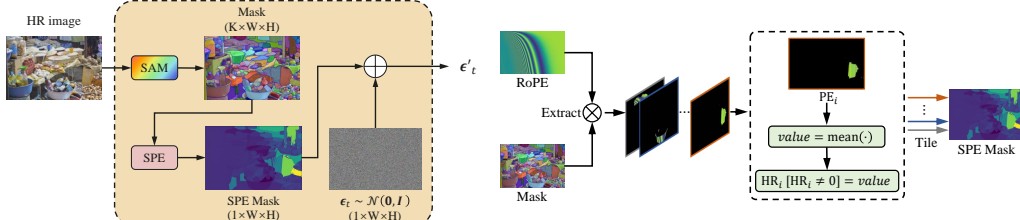

(a) Details of the structural noise modulation module.      (b) Details of the SPE module.

Figure 4: (a) During training, a SAM generates a segmentation mask for an HR image, and a structural position encoding (SPE) module encodes structure-level position information in the mask. The encoded mask is then added to the noise to modulate its mean in each segmentation area separately. At inference time, the framework utilizes only the trained diffusion model for image restoration, eliminating the inference cost of SAM. (b) This module encodes structural position information in the mask generated by SAM.

## 4 METHOD

### 4.1 OVERVIEW

In this paper, we present SAN-Diff, a structure-modulated diffusion framework designed to improve the performance of diffusion-based image SR models by leveraging fine-grained segmentation masks. As illustrated in Figure 3(b), these masks play a crucial role in a structural noise modulation module, modulating the mean of added noise in different segmentation areas during the forward process. Additionally, a structural position encoding (SPE) module is integrated to enrich the masks with structure-level position information.

We elaborate on the forward process in the proposed framework.[1] As discussed in Section 3.1, the added noise at each spatial point is independent and follows the same distribution, treating different areas in sample $\boldsymbol{x}_0$ equally during the forward process, even though they may possess different structural information and distributions. To address this limitation, we utilize a SAM to generate segmentation masks for modulating the added noise. The corresponding segmentation mask of $\boldsymbol{x}_0$ generated by SAM is denoted as $\boldsymbol{M}_{\text{SAM}}$. We then encode structural information into the mask using the SPE module, and the resulting encoded embedding mask is denoted as $\boldsymbol{E}_{\text{SAM}}$. Details of the SPE module will be provided in Section 4.2. At each step of the forward process, $\boldsymbol{E}_{\text{SAM}}$ is added to the standard Gaussian noise to inject structure-level information into the diffusion model. This modified process can be formulated as:

$$q(\boldsymbol{x}_t|\boldsymbol{x}_{t-1}, \boldsymbol{E}_{\text{SAM}}) = \mathcal{N}(\boldsymbol{x}_t; \sqrt{1 - \beta_t}\boldsymbol{x}_{t-1} + \sqrt{\beta_t}\boldsymbol{E}_{\text{SAM}}, \beta_t\mathbf{I}). \tag{2}$$

Compared with the original forward diffusion process defined in Equation 1, the modified process adds noise with different means to different segmentation areas. This makes local areas in an image distinguishable during forward diffusion, further aiding the diffusion model in learning a reverse process that makes more use of local information when generating an SR restoration for each area. Since the added Gaussian noise is independently sampled at each step, we can obtain the conditional distribution of $\boldsymbol{x}_t$ given $\boldsymbol{x}_0$ by iteratively applying the modified forward process:

$$q(\boldsymbol{x}_t|\boldsymbol{x}_0, \boldsymbol{E}_{\text{SAM}}) = \mathcal{N}(\boldsymbol{x}_t; \sqrt{\bar{\alpha}_t}\boldsymbol{x}_0 + \varphi_t\boldsymbol{E}_{\text{SAM}}, (1 - \bar{\alpha}_t)\mathbf{I}), \tag{3}$$

where $\alpha_t = 1 - \beta_t$, $\bar{\alpha}_t = \prod_{i=1}^{t} \alpha_i$, and $\varphi_t = \sum_{i=1}^{t} \sqrt{\bar{\alpha}_t \frac{\beta_i}{\alpha_i}}$. With this formula, we can directly derive the latent variable $\boldsymbol{x}_t$ from $\boldsymbol{x}_0$ in one step.

To achieve the SR image from restoration of an LR image, learning the reverse of the forward diffusion process is essential, characterized by the posterior distribution $p(\boldsymbol{x}_{t-1}|\boldsymbol{x}_t, \boldsymbol{E}_{\text{SAM}})$. However, the intractability arises due to the known marginal distributions $p(\boldsymbol{x}_{t-1})$ and $p(\boldsymbol{x}_t)$. This challenge is addressed by incorporating $\boldsymbol{x}_0$ into the condition. Employing Bayes' theorem, the posterior distribution $p(\boldsymbol{x}_{t-1}|\boldsymbol{x}_t, \boldsymbol{x}_0, \boldsymbol{E}_{\text{SAM}})$ can be formulated as:

---

[1]For additional details regarding the derivation, please refer to the supplementary material.

$$\tilde{\mu}_t(\boldsymbol{x}_t, \boldsymbol{x}_0, \boldsymbol{E}_{\text{SAM}}) = \frac{1}{\sqrt{\alpha_t}}(\boldsymbol{x}_t - \frac{\beta_t}{\sqrt{1-\bar{\alpha}_t}}(\frac{\sqrt{1-\bar{\alpha}_t}}{\sqrt{\beta_t}}\boldsymbol{E}_{\text{SAM}} + \boldsymbol{\epsilon})),$$

$$\tilde{\beta}_t = \frac{1-\bar{\alpha}_{t-1}}{1-\bar{\alpha}_t}\beta_t, \tag{4}$$

$$p(\boldsymbol{x}_{t-1}|\boldsymbol{x}_t, \boldsymbol{x}_0, \boldsymbol{E}_{\text{SAM}}) = \mathcal{N}(\boldsymbol{x}_{t-1}; \tilde{\mu}_t(\boldsymbol{x}_t, \boldsymbol{x}_0, \boldsymbol{E}_{\text{SAM}}), \tilde{\beta}_t\mathbf{I}),$$

where $\boldsymbol{\epsilon} \sim \mathcal{N}(0,1)$. To generate an SR image of an unseen LR image, we need to estimate the weighted summation of $\boldsymbol{E}_{\text{SAM}}$ and $\boldsymbol{\epsilon}$, as these variables are only defined in the forward process and cannot be accessed during inference. We adopt a denoising network $\boldsymbol{\epsilon_\theta}(\boldsymbol{x}_t, \boldsymbol{x}_{LR}, t)$ for approximation. The associated loss function is formulated as:

$$\mathcal{L}(\boldsymbol{\theta}) = \mathbb{E}_{t, \boldsymbol{x}_0, \boldsymbol{\epsilon}}[\|\frac{\sqrt{1-\bar{\alpha}_t}}{\sqrt{\beta_t}}\boldsymbol{E}_{\text{SAM}} + \boldsymbol{\epsilon} - \boldsymbol{\epsilon_\theta}(\boldsymbol{x}_t, \boldsymbol{x}_{LR}, t)\|_2^2]. \tag{5}$$

The denoising network $\boldsymbol{\epsilon_\theta}(\boldsymbol{x}_t, \boldsymbol{x}_{LR}, t)$ predicts the weighted summation based on latent variable $\boldsymbol{x}_t$, LR image $\boldsymbol{x}_{LR}$, and step $t$. During training, $\boldsymbol{x}_t$ is derived by sampling from the distribution defined in Equation 3. At inference time, the restored sample at step $t$ is used as $\boldsymbol{x}_t$.

**Discussion.** The structure-level information encoded by the mask can be injected into the diffusion model through two distinct approaches. One method involves using the mask to modulate the input of the diffusion model, while the other method entails modulating the noise in the forward process, which is the approach adopted in our proposed method. In comparison to directly modulating the input, our method only requires the oracle SAM during training. Subsequently, the trained diffusion model can independently restore the SR image of an unseen LR image by iteratively applying the posterior distribution defined in Equation 4. This highlights that our SAN-Diff method incurs *no additional inference cost* during inference.

## 4.2 STRUCTURAL POSITION ENCODING

After obtaining the original segmentation mask using SAM, we employ an SPE module to encode structural position information in the mask. Details of this module are illustrated in Figure 4(b).

The fundamental concept behind the SPE module is to assign a unique value to each segmentation area. The segmentation mask generated by SAM comprises a series of 0-1 masks, where each mask corresponds to an area in the original image sharing the same semantic information. Consequently, for HR image $\boldsymbol{x}_{HR}^{3\times h\times w}$, we can represent the $K$ segmentation masks as $\boldsymbol{M}_{\text{SAM},i}$, where $i = 1, 2, \cdots, K$ is the index of different areas in the original image. Specifically, the value of a point in $\boldsymbol{M}_{\text{SAM},i} \in {0,1}^{1\times h\times w}$ equals 1 when its position is within the $i$-th area in the original image and 0 otherwise. To encode position information, we generate a rotary position embedding (RoPE) (Su et al., 2021) $\boldsymbol{x}_{\text{RoPE}} \in \mathbb{R}^{1\times h\times w}$, where the width is considered the length of the sequence and the height is considered the embedding dimension in RoPE. We initialize $\boldsymbol{x}_{\text{RoPE}}$ with a 1-filled tensor Similarly, $\boldsymbol{x}_{\text{RoPE}}$ can be decomposed as: $\boldsymbol{x}_{\text{RoPE}} = \sum_i \boldsymbol{x}_{\text{RoPE},i} = \sum_i \boldsymbol{x}_{\text{RoPE}} \cdot \boldsymbol{M}_{\text{SAM},i}$. Subsequently, we obtain the structurally positioned embedded mask by:

$$\boldsymbol{E}_{\text{SAM}} = \sum_i \boldsymbol{M}_{\text{SAM},i} \cdot \text{mean}(\boldsymbol{x}_{\text{RoPE},i}), \tag{6}$$

which means to assign the average value of $\boldsymbol{x}_{\text{RoPE},i}$ to $i$-th segmentation area.

## 4.3 TRAINING AND INFERENCE

The training of the diffusion model necessitates segmentation masks for all HR images in the training set. We employ SAM to generate these masks. This process is executed once before training, and the generated masks are reused in all epochs. Therefore, our method incurs only a negligible additional training cost from the integration of SAM. Subsequently, a model is trained to estimate the modulated noise in the forward diffusion process using the loss function outlined in Equation 5.

During inference, we follow the practice of SRDiff(Li et al., 2022), the restoration of SR images can be accomplished by applying the reverse diffusion process to LR images. By iteratively applying the posterior distribution in Equation 4 and utilizing the trained model to estimate the mean, the restoration of the corresponding SR image is achieved. It is noteworthy that we opted for the $x_T$

Table 2: Results on test sets of several public benchmarks and the validation set of DIV2K. We report the results achieved by GAN-based and diffusion-based methods. (↑) and (↓) indicate that a larger or smaller corresponding score is better, respectively. Best and second best performance are in red and blue colors, respectively.

| Method | Urban100 | | | | BSDS100 | | | | Manga109 | | | | General100 | | | | DIV2K | | | |
|---|---|---|---|---|---|---|---|---|---|---|---|---|---|---|---|---|---|---|---|---|
| | PSNR (↑) | SSIM (↑) | MANIQA (↑) | FID (↓) | PSNR (↑) | SSIM (↑) | MANIQA (↑) | FID (↓) | PSNR (↑) | SSIM (↑) | MANIQA (↑) | FID (↓) | PSNR (↑) | SSIM (↑) | MANIQA (↑) | FID (↓) | PSNR (↑) | SSIM (↑) | MANIQA (↑) | FID (↓) |
| SRGAN | 22.85 | 0.6846 | 0.6162 | 10.4991 | 24.75 | 0.6400 | 0.6058 | 54.50 | 28.08 | 0.8616 | 0.5822 | 4.1818 | 25.98 | 0.7470 | 0.6172 | 36.23 | 28.05 | 0.7738 | 0.5600 | 2.0889 |
| SFTGAN | 21.95 | 0.6457 | 0.6153 | 9.1382 | 24.69 | 0.6365 | 0.6173 | 49.30 | 20.72 | 0.7008 | 0.5687 | 9.6466 | 22.19 | 0.6432 | 0.6253 | 37.06 | 24.70 | 0.6929 | 0.5602 | 5.1979 |
| ESRGAN | 22.99 | 0.6940 | 0.6678 | 7.3793 | 24.65 | 0.6374 | 0.6449 | 45.88 | 28.60 | 0.8553 | 0.6026 | 3.1242 | 26.03 | 0.7449 | 0.6452 | 30.93 | 28.18 | 0.7709 | 0.5849 | 1.4586 |
| USRGAN | 23.23 | 0.7060 | 0.6785 | 6.4375 | 25.13 | 0.6604 | 0.6517 | 48.58 | 20.70 | 0.7092 | 0.6226 | 8.6123 | 26.35 | 0.7631 | 0.6411 | 35.22 | 28.79 | 0.7945 | 0.5827 | 0.5938 |
| SPSR | 23.05 | 0.6973 | 0.6823 | 7.8380 | 24.60 | 0.6375 | 0.6648 | 48.81 | 23.26 | 0.7740 | 0.6211 | 6.6369 | 25.96 | 0.7435 | 0.6571 | 30.94 | 28.19 | 0.7727 | 0.5945 | 1.4315 |
| BSRGAN | 22.37 | 0.6628 | 0.6334 | 33.7447 | 24.95 | 0.6365 | 0.5993 | 114.08 | 26.09 | 0.8272 | 0.6105 | 33.5110 | 25.23 | 0.7309 | 0.6337 | 86.14 | 27.32 | 0.7577 | 0.5616 | 14.1312 |
| LDM | 22.23 | 0.6546 | 0.6239 | 23.0718 | 23.56 | 0.5812 | 0.6194 | 109.77 | 24.26 | 0.7941 | 0.5870 | 20.7506 | 25.32 | 0.6779 | 0.5683 | 265.82 | 26.45 | 0.7340 | 0.5356 | 9.5518 |
| StableSR | 21.16 | 0.6529 | 0.7025 | 28.9426 | 24.64 | 0.6523 | 0.6606 | 68.77 | 21.22 | 0.7456 | 0.6576 | 31.4120 | 18.39 | 0.5324 | 0.6749 | 73.51 | 26.83 | 0.7653 | 0.5747 | 14.5232 |
| StableSR(Turbo) | 21.22 | 0.6658 | 0.6633 | 29.5486 | 24.61 | 0.6691 | 0.6347 | 74.04 | 22.68 | 0.7819 | 0.5875 | 29.1558 | 18.63 | 0.5421 | 0.6446 | 67.04 | 26.68 | 0.7776 | 0.5468 | 14.2138 |
| DiffBIR | 22.40 | 0.6417 | 0.6536 | 30.6352 | 25.09 | 0.6254 | 0.6626 | 69.18 | 21.81 | 0.7197 | 0.6251 | 30.6433 | 24.37 | 0.6878 | 0.6762 | 66.35 | 26.25 | 0.7051 | 0.5919 | 17.8206 |
| SRDiff | 25.08 | 0.7602 | 0.6604 | 5.2194 | 25.86 | 0.6805 | 0.6478 | 56.27 | 28.78 | 0.8764 | 0.5967 | 2.4929 | 29.82 | 0.8223 | 0.6500 | 36.35 | 28.60 | 0.7908 | 0.5910 | 0.4649 |
| SAN-Diff (Ours) | 25.54 | 0.7721 | 0.6709 | 4.5276 | 26.47 | 0.7003 | 0.6667 | 60.81 | 29.43 | 0.8899 | 0.6046 | 2.3994 | 30.30 | 0.8353 | 0.6346 | 38.42 | 29.34 | 0.8109 | 0.5959 | 0.3809 |

sample from $\mathcal{N}(0, \mathbf{I})$ instead of $\mathcal{N}(\varphi_T \boldsymbol{E}_{\text{SAM}}, \mathbf{I})$. Because the denoising model can generate the correct noise distribution, the initial distribution is not expected to exert a significant impact on the ultimately reconstructed image during the iterative denoising process. Simultaneously, such choice also ensures that our SAN-Diff method without additional inference cost.

## 5 EXPERIMENT

### 5.1 EXPERIMENTAL SETUP

**Dataset.** We evaluate the proposed method on the general SR (4×) task. The training data in DIV2K (Agustsson & Timofte, 2017) and all data in Flickr2K (Wang et al., 2019) are adopted as the training set. For images in the training set, we adopt a SAM to obtain their corresponding segmentation masks. After that, structural position information is encoded into the mask by the SPE module in our proposed framework. We adopt a patch size settings of 160×160 to crop each image and its corresponding mask. For evaluation, several commonly-used SR testing dataset are used, including Set14 (Zeyde et al., 2012), Urban100 (Huang et al., 2015), BSDS100 (Martin et al., 2001), Manga109 (Fujimoto et al., 2016), General100 (Dong et al., 2016). Besides, the validation set of DIV2K (Agustsson & Timofte, 2017) is also used for evaluation.

**Baseline.** We choose a wide range of methods for comparison. Among them, SRGAN (Ledig et al., 2017), SFTGAN (Wang et al., 2018a), ESRGAN (Wang et al., 2018b), BSRGAN (Zhang et al., 2021), USRGAN (Zhang et al., 2020), and SPSR (Ma et al., 2020) are GAN-based generative methods. Besides, we also comparison with diffusion-base generative methods such as LDM (Rombach et al., 2022b), StableSR (Wang et al., 2023), DiffBIR (Lin et al., 2023), and SRDiff (Li et al., 2022), .

**Model architecture.** Architecture of the used denoising model in our experiments follows Li *et al.* (Li et al., 2022). As for configuration of the forward diffusion process, we set the number of diffusing steps $T$ to 100 and scheduling hyperparameters $\beta_1, \ldots, \beta_T$ following Nichol *et al.* (Nichol & Dhariwal, 2021)

**Optimization.** We train the diffusion model for 400K iterations with a batch size of 16, and adopt Adam (Kingma & Ba, 2014) as the optimizer. The initial learning rate is $2 \times 10^{-4}$ and the cosine learning rate decay is adopted. The training process requires approximately 75 hours and 30GB of GPU memory on a single GPU card.

**Metric.** Both objective and subjective metrics are used in our experiment. PSNR and SSIM (Wang et al., 2004) serve as objective metrics for quantitative measurements, which are computed over the Y-channel after converting SR images from the RGB space to the YUV space. To evaluate the perceptual quality, we also adopt Fréchet inception distance (FID) (Heusel et al., 2017) and MANIQA (Yang et al., 2022) as the subjective metric, which measures the fidelity and diversity of generated images.

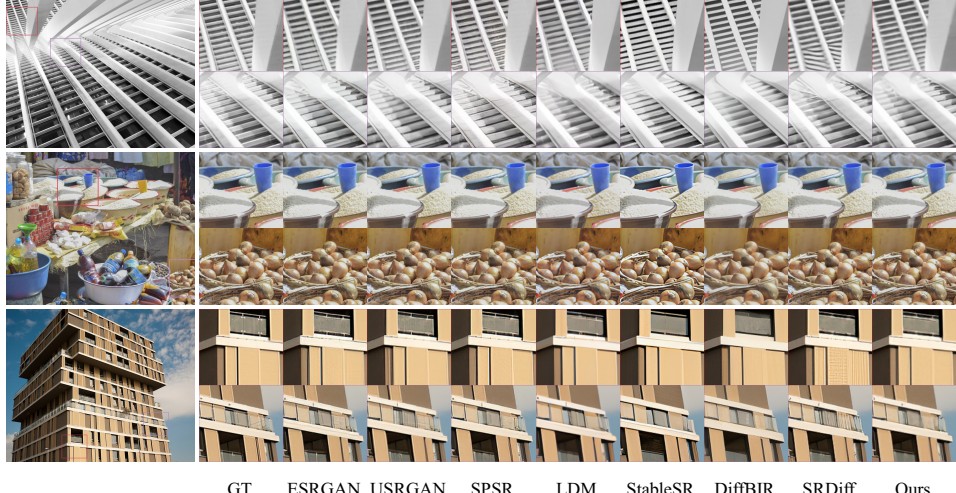

GT   ESRGAN  USRGAN  SPSR   LDM   StableSR  DiffBIR  SRDiff   Ours

Figure 5: Visualization of restored images generated by different methods. Our SAN-Diff surpasses other approaches in terms of both higher reconstruction quality and fewer artifacts. Additional visualization results can be found in our supplementary material.

## 5.2 PERFORMANCE OF IMAGE SR

We compare the performance of the proposed SAN-Diff method with baselines on several commonly used benchmarks for image SR. The quantitative results are presented in Table 2. In the results, our method outperforms the diffusion-based baseline SRDiff in terms of all three metrics, except a slightly higher FID score on BSDS100 and General100. Moreover, SAN-Diff can even achieves better performance when compared to conventional approaches.

Figure 5 presents several images by generated different methods. Compared with the baselines, our methods is able to generate more realistic details of the given image. Moreover, the reconstructed images contain less artifact, which refers to the unintended distortion or anomalies in the SR image. We further evaluate the proposed method in terms of inhibiting artifact in Section 5.3.

Table 3: Averaged value of artifact maps. Lower value indicates fewer artifacts in SR images.

| Method | Set5 | Set14 | Urban100 | BSDS100 | Manga109 | General100 | DIV2K |
|---|---|---|---|---|---|---|---|
| SRGAN (Ledig et al., 2017) | 0.2263 | 1.3248 | 2.7320 | 1.2158 | 0.4736 | 1.4216 | 0.7456 |
| SFTGAN (Wang et al., 2018a) | 0.9014 | 2.0866 | 4.4362 | 1.2137 | 5.7064 | 3.6220 | 2.1495 |
| ESRGAN (Wang et al., 2018b) | 0.1842 | 1.4140 | 2.7006 | 1.2331 | 0.4042 | 1.4331 | 0.7335 |
| USRGAN (Zhang et al., 2020) | 0.1661 | 1.1537 | 2.5297 | 1.0947 | 5.7367 | 1.3029 | 0.6239 |
| SPSR (Ma et al., 2020) | 0.1653 | 1.3096 | 2.7069 | 1.2467 | 2.6665 | 1.4701 | 0.7295 |
| BSRGAN (Zhang et al., 2021) | 0.5255 | 1.3557 | 2.9030 | 1.1467 | 1.0150 | 1.5147 | 0.7718 |
| LDM (Rombach et al., 2022b) | 0.7735 | 2.1252 | 3.4932 | 1.8173 | 1.9994 | 1.5201 | 1.0334 |
| StableSR (Wang et al., 2023) | 3.4917 | 5.6209 | 4.0859 | 1.2014 | 4.5033 | 8.4946 | 0.8749 |
| StableSR(Turbo) (Wang et al., 2023) | 3.1433 | 5.4212 | 3.9131 | 1.2598 | 2.9956 | 8.3225 | 0.8883 |
| DiffBIR (Lin et al., 2023) | 0.9508 | 1.5292 | 2.5967 | 1.0446 | 4.0051 | 1.8129 | 1.0440 |
| SRDiff (Li et al., 2022) | 0.1821 | 0.7375 | 1.4163 | 1.2226 | 0.4047 | 0.4370 | 0.6185 |
| SAN-Diff(Ours) | **0.1322** | **0.5804** | **1.1453** | **0.9226** | **0.3081** | **0.3145** | **0.4391** |

## 5.3 PERFORMANCE OF INHIBITING ARTIFACT

Generative image SR models excel at recovering sharp images with rich details. However, they are prone to unintended distortions or anomalies in the restored images (Liang et al., 2022a), commonly referred to as artifacts. In our experiments, we closely examine the performance of our method in inhibiting artifacts.

Following the approach outlined in (Liang et al., 2022a), we calculate the artifact map for each SR image. Table 3 presents the averaged values of artifact maps on four datasets, and Figure 6 visually showcases the artifact maps. When compared with other methods, our SAN-Diff demonstrates the ability to generate SR images with fewer artifacts, as supported by both quantitative and qualitative assessments.

ESRGAN    USRGAN    SPSR    LDM    StableSR    DiffBIR    SRDiff    Ours

Figure 6: Visualization of artifact maps. Bright regions indicate artifacts in the restored images. Our proposed method generates images with fewer artifacts compared to other methods.

## 5.4 ABLATION STUDY

**Quality of mask.** Segmentation masks provide the diffusion model structure-level information during training. We conduct experiments to study the impact of using masks with different qualities. Specifically, masks with three qualities are considered: those that are generated by Mobile-SAM (Zhang et al., 2023) using LR images, those that are generated by MobileSAM using HR images, and those that are generated the original SAM (Kirillov et al., 2023) using HR images. These three kinds of masks are referred to as "Low", "Medium", and "High", respectively. The results of comparing masks with varying qualities are presented in Table 4, indicating that the final performance of the trained model improves as the mask quality increases on both the Urban100 and DIV2k datasets. These findings demonstrate the critical role of high-quality masks in achieving exceptional performance.

**Structural position embedding.** In our SPE module, the RoPE is adopted to generate a 2D position embedding map for obtaining the value assigned to each segmentation area. Here we consider two other approaches: one is using a cosine function to generate a 2D grid as the position embedding map, and the other one is using a linear function whose output value ranges from 0 to 1 to generate the 2D grid. Table 5 shows the corresponding results. Compared with using 2D grids generated with cosine or linear functions, utilizing that generated by RoPE to calculate the value assigned to each segmentation area results in superior performance, thereby showcasing the effectiveness of our SPE module design.

Table 4: Comparison of masks with different qualities.

| Mask quality | Urban100 | | | DIV2K | | |
|---|---|---|---|---|---|---|
| | PSNR (↑) | SSIM (↑) | FID (↓) | PSNR (↑) | SSIM (↑) | FID (↓) |
| Low | 25.33 | 0.7702 | 4.7100 | 29.09 | 0.8062 | 0.4480 |
| Medium | 25.40 | 0.7700 | 4.7576 | 29.30 | 0.8103 | 0.4176 |
| High | **25.54** | **0.7721** | **4.5276** | **29.34** | **0.8109** | **0.3809** |

Table 5: Comparison of different schemes for position embedding.

| Position embedding | Urban100 | | | DIV2K | | |
|---|---|---|---|---|---|---|
| | PSNR (↑) | SSIM (↑) | FID (↓) | PSNR (↑) | SSIM (↑) | FID (↓) |
| Cosine | 25.28 | 0.7670 | 4.7790 | 28.98 | 0.8033 | 0.4689 |
| Linear | 25.31 | 0.7693 | 4.6197 | 29.09 | 0.8073 | 0.4731 |
| RoPE | **25.54** | **0.7721** | **4.5276** | **29.34** | **0.8109** | **0.3809** |

## 6 CONCLUSION

This paper focuses on enhancing the structure-level information restoration capability of diffusion-based image SR models through the integration of SAM. Specifically, we introduce a framework named SAN-Diff, which involves the incorporation of structural position information into the SAM-generated mask, followed by its addition to the sampled noise during the forward diffusion process. This operation individually modulates the mean of the noise in each corresponding segmentation area, thereby injecting structure-level knowledge into the diffusion model. Through the adoption of this method, trained model demonstrates an improvement in the restoration of structural details and the inhibition of artifacts in images, all without incurring any additional inference cost. The effectiveness of our method is substantiated through extensive experiments conducted on commonly used image SR benchmarks.

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

## A  ALGORITHM DETAILS

Here we provide algorithm details of our SAN-Diff framework. We adopt the original notations in denoising diffusion probabilistic model (DDPM) (Ho et al., 2020b). Given a data sample $\boldsymbol{x}_0 \in p_{\text{data}}$, the proposed framework in DDPM is defined as:

$$q(\boldsymbol{x}_t|\boldsymbol{x}_{t-1}) = \mathcal{N}(\boldsymbol{x}_t; \sqrt{1 - \beta_t}\boldsymbol{x}_{t-1}, \beta_t\mathbf{I}), \tag{7}$$

where $\boldsymbol{x}_t$ is the noise latent variable at step $t$. $\beta_1, \ldots, \beta_T \in (0, 1)$ are hyperparameters scheduling the scale of added noise for $T$ steps. Given $\boldsymbol{x}_{t-1}$ We can sample $\boldsymbol{x}_t$ from this distribution by:

$$\boldsymbol{x}_t = \sqrt{1 - \beta_t}\boldsymbol{x}_{t-1} + \sqrt{\beta_t}\boldsymbol{\epsilon}_t, \tag{8}$$

where $\boldsymbol{\epsilon}_t \sim \mathcal{N}(0, \mathbf{I})$.

In our SAN-Diff framework, we use a structural position encoded segmentation mask $\boldsymbol{E}_{\text{SAM}}$ to modulate the standard Gaussian noise used in the original DDPM by adding $\boldsymbol{E}_{\text{SAM}}$ to $\boldsymbol{\epsilon}_t$. Then the sampling of $\boldsymbol{x}_t$ becomes:

$$\boldsymbol{x}_t = \sqrt{1 - \beta_t}\boldsymbol{x}_{t-1} + \sqrt{\beta_t}(\boldsymbol{\epsilon}_t + \boldsymbol{E}_{\text{SAM}}), \tag{9}$$

and its corresponding conditional distribution is:

$$q(\boldsymbol{x}_t|\boldsymbol{x}_{t-1}, \boldsymbol{E}_{\text{SAM}}) = \mathcal{N}(\boldsymbol{x}_t; \sqrt{1 - \beta_t}\boldsymbol{x}_{t-1} + \sqrt{\beta_t}\boldsymbol{E}_{\text{SAM}}, \beta_t\mathbf{I}). \tag{10}$$

Let $\alpha_t = 1 - \beta_t$ and iteratively apply Equation 9, we have:

$$
\begin{aligned}
\boldsymbol{x}_t &= \sqrt{\alpha_t}(\sqrt{\alpha_{t-1}}(\ldots) + \sqrt{\beta_{t-1}}\boldsymbol{E}_{\text{SAM}} + \sqrt{\beta_{t-1}}\boldsymbol{\epsilon}_{t-1}) + \sqrt{\beta_t}\boldsymbol{E}_{\text{SAM}} + \sqrt{\beta_t}\boldsymbol{\epsilon}_t \\
&= \sqrt{\alpha_t \ldots \alpha_1}\boldsymbol{x}_0 + (\sqrt{\alpha_t \ldots \alpha_2\beta_1} + \cdots + \sqrt{\beta_t})\boldsymbol{E}_{\text{SAM}} + (\sqrt{\alpha_t \ldots \alpha_2\beta_1}\boldsymbol{\epsilon}_1 + \cdots + \sqrt{\beta_t}\boldsymbol{\epsilon}_t) \\
&= \sqrt{\bar{\alpha}_t}\boldsymbol{x}_0 + \varphi_t\boldsymbol{E}_{\text{SAM}} + \sqrt{1 - \bar{\alpha}_t}\boldsymbol{\epsilon},
\end{aligned}
\tag{11}
$$

where $\bar{\alpha}_t = \prod_{i=1}^t \alpha_i$, $\varphi_t = \sqrt{\alpha_t \ldots \alpha_2\beta_1} + \cdots + \sqrt{\beta_t} = \sum_{i=1}^t \sqrt{\bar{\alpha}_t \frac{\beta_i}{\bar{\alpha}_i}}$, and $\boldsymbol{\epsilon} \sim \mathcal{N}(0, \mathbf{I})$.

The corresponding conditional distribution is:

$$q(\boldsymbol{x}_t|\boldsymbol{x}_0, \boldsymbol{E}_{\text{SAM}}) = \mathcal{N}(\boldsymbol{x}_t; \sqrt{\bar{\alpha}_t}\boldsymbol{x}_0 + \varphi_t\boldsymbol{E}_{\text{SAM}}, (1 - \bar{\alpha}_t)\mathbf{I}). \tag{12}$$

Then similar to the original DDPM, we are interested in the posterior distribution that defines the reverse diffusion process. With Bayes' theorem, it can be formulated as:

$$
\begin{aligned}
p(\boldsymbol{x}_{t-1}|\boldsymbol{x}_t, \boldsymbol{x}_0, \boldsymbol{E}_{\text{SAM}}) &= \frac{p(\boldsymbol{x}_t|\boldsymbol{x}_{t-1})p(\boldsymbol{x}_{t-1}|\boldsymbol{x}_0, \boldsymbol{E}_{\text{SAM}})}{p(\boldsymbol{x}_t|\boldsymbol{x}_0, \boldsymbol{E}_{\text{SAM}})} \\
&\propto \exp\left(-\frac{1}{2}\left(\left(\frac{\alpha_t}{\beta_t} + \frac{1}{1 - \bar{\alpha}_{t-1}}\right)\boldsymbol{x}_{t-1}^2 - 2\left(\frac{\sqrt{\alpha_t}(\boldsymbol{x}_t - \sqrt{\beta_t}\boldsymbol{E}_{\text{SAM}})}{\beta_t} + \frac{\sqrt{\bar{\alpha}_{t-1}}\boldsymbol{x}_0 + \varphi_{t-1}\boldsymbol{E}_{\text{SAM}}}{1 - \bar{\alpha}_{t-1}}\right)\boldsymbol{x}_{t-1}\right)\right) \\
&\quad + C(\boldsymbol{x}_t, \boldsymbol{x}_0, \boldsymbol{E}_{\text{SAM}}),
\end{aligned}
\tag{13}
$$

where $C(\boldsymbol{x}_t, \boldsymbol{x}_0, \boldsymbol{E}_{\text{SAM}})$ not involves $\boldsymbol{x}_{t-1}$. The posterior is also a Gaussian distribution. By using the following notations:

$$\tilde{\beta}_t = 1 \Big/ \left(\frac{\alpha_t}{\beta_t} + \frac{1}{1 - \bar{\alpha}_{t-1}}\right) = \frac{1 - \bar{\alpha}_{t-1}}{1 - \bar{\alpha}_t}\beta_t, \tag{14}$$

$$
\begin{aligned}
\tilde{\mu}_t(\boldsymbol{x}_t, \boldsymbol{x}_0, \boldsymbol{E}_{\text{SAM}}) &= \left(\frac{\sqrt{\alpha_t}(\boldsymbol{x}_t - \sqrt{\beta_t}\boldsymbol{E}_{\text{SAM}})}{\beta_t} + \frac{\sqrt{\bar{\alpha}_{t-1}}\boldsymbol{x}_0 + \varphi_{t-1}\boldsymbol{E}_{\text{SAM}}}{1 - \bar{\alpha}_{t-1}}\right) \cdot \tilde{\beta}_t \\
&= \frac{1}{\sqrt{\alpha_t}}\left(\boldsymbol{x}_t - \frac{\beta_t}{\sqrt{1 - \bar{\alpha}_t}}\left(\frac{\sqrt{1 - \bar{\alpha}_t}}{\sqrt{\beta_t}}\boldsymbol{E}_{\text{SAM}} + \boldsymbol{\epsilon}\right)\right),
\end{aligned}
\tag{15}
$$

the posterior distribution can be formulated as:

$$p(\boldsymbol{x}_{t-1}|\boldsymbol{x}_t, \boldsymbol{x}_0, \boldsymbol{E}_{\text{SAM}}) = \mathcal{N}(\boldsymbol{x}_{t-1}; \tilde{\mu}_t(\boldsymbol{x}_t, \boldsymbol{x}_0, \boldsymbol{E}_{\text{SAM}}), \tilde{\beta}_t\mathbf{I}). \tag{16}$$

Given latent variable $\boldsymbol{x}_t$, we want to sample from the posterior distribution to obtain the denoised latent variable $\boldsymbol{x}_{t-1}$. This requires the estimation of $\tilde{\mu}_t(\boldsymbol{x}_t, \boldsymbol{x}_0, \boldsymbol{E}_{\mathrm{SAM}})$, *i.e.*, the estimation of $\frac{\sqrt{1-\bar{\alpha}_t}}{\sqrt{\beta_t}}\boldsymbol{E}_{\mathrm{SAM}} + \boldsymbol{\epsilon}$. This is achieved by a parameterized denoising network $\boldsymbol{\epsilon}_{\boldsymbol{\theta}}(\boldsymbol{x}_t, t)$. The loss function is:

$$
\begin{aligned}
\mathcal{L}(\boldsymbol{\theta}) &= \mathbb{E}_{t,\boldsymbol{x}_0,\boldsymbol{\epsilon}}[\|\frac{\sqrt{1-\bar{\alpha}_t}}{\sqrt{\beta_t}}\boldsymbol{E}_{\mathrm{SAM}} + \boldsymbol{\epsilon} - \boldsymbol{\epsilon}_{\boldsymbol{\theta}}(\boldsymbol{x}_t, t)\|_2^2] \\
&= \mathbb{E}_{t,\boldsymbol{x}_0,\boldsymbol{\epsilon}}[\|\frac{\sqrt{1-\bar{\alpha}_t}}{\sqrt{\beta_t}}\boldsymbol{E}_{\mathrm{SAM}} + \boldsymbol{\epsilon} - \boldsymbol{\epsilon}_{\boldsymbol{\theta}}(\sqrt{\bar{\alpha}_t}\boldsymbol{x}_0 + \varphi_t \boldsymbol{E}_{\mathrm{SAM}} + \sqrt{1-\bar{\alpha}_t}\boldsymbol{\epsilon}, t)\|_2^2].
\end{aligned}
\tag{17}
$$

This is the loss function in our main paper. Note that the form of $\tilde{\mu}_t(\boldsymbol{x}_t, \boldsymbol{x}_0, \boldsymbol{E}_{\mathrm{SAM}})$ is same to that in the original DDPM. Therefore, our framework requires no change of the generating process and brings no additional inference cost.

## B  ABLATION STUDY

**Non-informative segmentation mask.** There are cases where all pixels in a training sample belongs to the same segmentation area because of the patch-splitting scheme used during training. Two schemes are considered to cope with such non-informative segmentation mask: directly using the original mask, or adopting a special mask filled with fixed values, *i.e.*, zeros. Table 6 presents the comparison results of the above two schemes. Based on the results, it is advantageous to convert non-informative segmentation masks into an all-zero matrix. Our speculation is that the model may be confused by various values in non-informative segmentation masks across different samples, if no reduction is applied to unify such scenarios.

Table 6: Comparison of two schemes for handling non-informative masks. "Reduce" indicates that the mask is replaced with a zero-filled matrix when all pixels belong to the same segmentation area. Otherwise, the original mask is used.

| Reduce | Urban100 | | | DIV2K | | |
|---|---|---|---|---|---|---|
| | PSNR ($\uparrow$) | SSIM ($\uparrow$) | FID ($\downarrow$) | PSNR ($\uparrow$) | SSIM ($\uparrow$) | FID ($\downarrow$) |
| ✗ | 25.40 | 0.7687 | 4.7149 | 29.18 | 0.8064 | 0.4673 |
| ✓ | **25.54** | **0.7721** | **4.5276** | **29.34** | **0.8109** | **0.3809** |

**Model performance at different super-resolution scales.** We conducted experiments on the X2 setting, and the results show that our method has a significant performance improvement over the baseline on the reference metric, while maintaining the same level on the unreferenced metric.

Table 7: X2 scale results on test sets of several public benchmarks. ($\uparrow$) and ($\downarrow$) indicate that a larger or smaller corresponding score is better, respectively.

| Method | Urban100 | | | | BSDS100 | | | | Manga109 | | | | General100 | | | | DIV2K | | | |
|---|---|---|---|---|---|---|---|---|---|---|---|---|---|---|---|---|---|---|---|---|
| | PSNR ($\uparrow$) | SSIM ($\uparrow$) | MANIQA ($\uparrow$) | FID ($\downarrow$) | PSNR ($\uparrow$) | SSIM ($\uparrow$) | MANIQA ($\uparrow$) | FID ($\downarrow$) | PSNR ($\uparrow$) | SSIM ($\uparrow$) | MANIQA ($\uparrow$) | FID ($\downarrow$) | PSNR ($\uparrow$) | SSIM ($\uparrow$) | MANIQA ($\uparrow$) | FID ($\downarrow$) | PSNR ($\uparrow$) | SSIM ($\uparrow$) | MANIQA ($\uparrow$) | FID ($\downarrow$) |
| SRDiff (X2) | 30.84 | 0.9080 | 0.5265 | 0.2067 | 36.87 | 0.9667 | 0.4176 | 0.0679 | 30.05 | 0.8541 | 0.4545 | 10.2967 | 36.43 | 0.9431 | 0.4852 | 6.2866 | 34.05 | 0.9178 | 0.3853 | 0.0292 |
| SAN-Diff (X2) | 30.88 | 0.9095 | 0.5246 | 0.2145 | 37.08 | 0.9679 | 0.4192 | 0.0692 | 30.36 | 0.8628 | 0.4346 | 10.4271 | 36.69 | 0.9458 | 0.4824 | 6.4630 | 34.33 | 0.9230 | 0.3832 | 0.0287 |

## C  DISCUSSION

### C.1  EXTENSION TO OTHER DIFFUSION TASKS

Our framework has the flexibility to accommodate such tasks seamlessly, as the SAM information functions like a plugin without necessitating alterations to the original diffusion framework. Previous works [1] have demonstrated the efficacy of diffusion-based frameworks across various

low-level tasks such as inpainting and deblurring. We are confident that our framework can similarly excel in these areas. However, it's worth noting that our method modifies the diffusion process, which means that simple fine-tuning of pretrained models using parameter efficient approaches like LoRA is not suitable. Instead, retraining the model becomes necessary, which poses computational challenges due to resource constraints. Given these limitations, our paper primarily focuses on the image SR task. Nonetheless, we are committed to expanding our method to encompass a broader range of tasks in the future. We eagerly anticipate collaboration with the computer vision community to further explore these possibilities.

## C.2 REALISTIC FINE-GRAINED TEXTURES

In the field of Image SR, models sometimes generate images with seemingly fine-grained textures, even though the LR images do not contain recognizable textures to the human eye. Defining the correctness of generated texture in such cases presents a challenge. In addressing this issue, we believe that exploring how to generate realistic fine-grained textures within our framework by integrating other kinds of prior information into the model would be a valuable research direction.

## C.3 LIMITATION FROM THE ABILITY OF THE SEGMENTATION MODEL

Compared to the original diffusion model without structural guidance, masks generated by existing SAM models can improve performance, as demonstrated in our experimental results.

However, the performance of our model does depend on the quality of the segmentation masks, as they capture the structural information of the corresponding image. Our model benefits from SAM's fine-grained segmentation capability and its strong generalization ability across diverse objects and textures in the real world. Nevertheless, the performance of our model is also limited by the capabilities of the segmentation model itself. For instance, SAM may struggle to identify structures with low resolution in certain scenes. While the model can partially mitigate this issue by learning from a large amount of data during training, it is undeniable that higher segmentation precision (e.g., SAM2) and finer segmentation granularity would significantly enhance the performance of our approach.

## C.4 SOCIETAL IMPACT

Although our work focuses on improving the performance of diffusion models in super-resolution tasks, the proposed framework can be applied to any task based on diffusion models. This may result in generative models producing higher-quality and more difficult-to-detect deepfakes.

## C.5 SAM INFERENCE RESULT VISUALIZATION

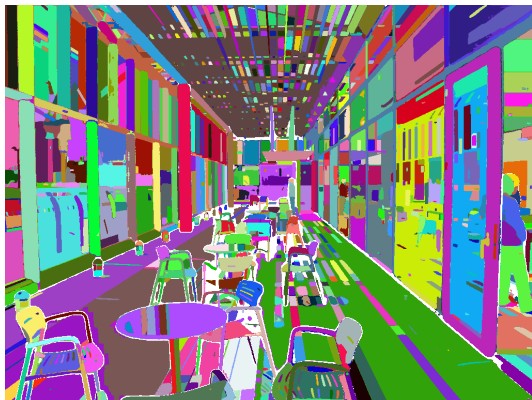

Figure 7: We visualized the results obtained by applying SAM inference to the original images in Figure 1(B). These results are not involved in the inference process. It is only used as a reference for analyzing the super-resolution result.

## D MULTI-METRIC COMPARISON

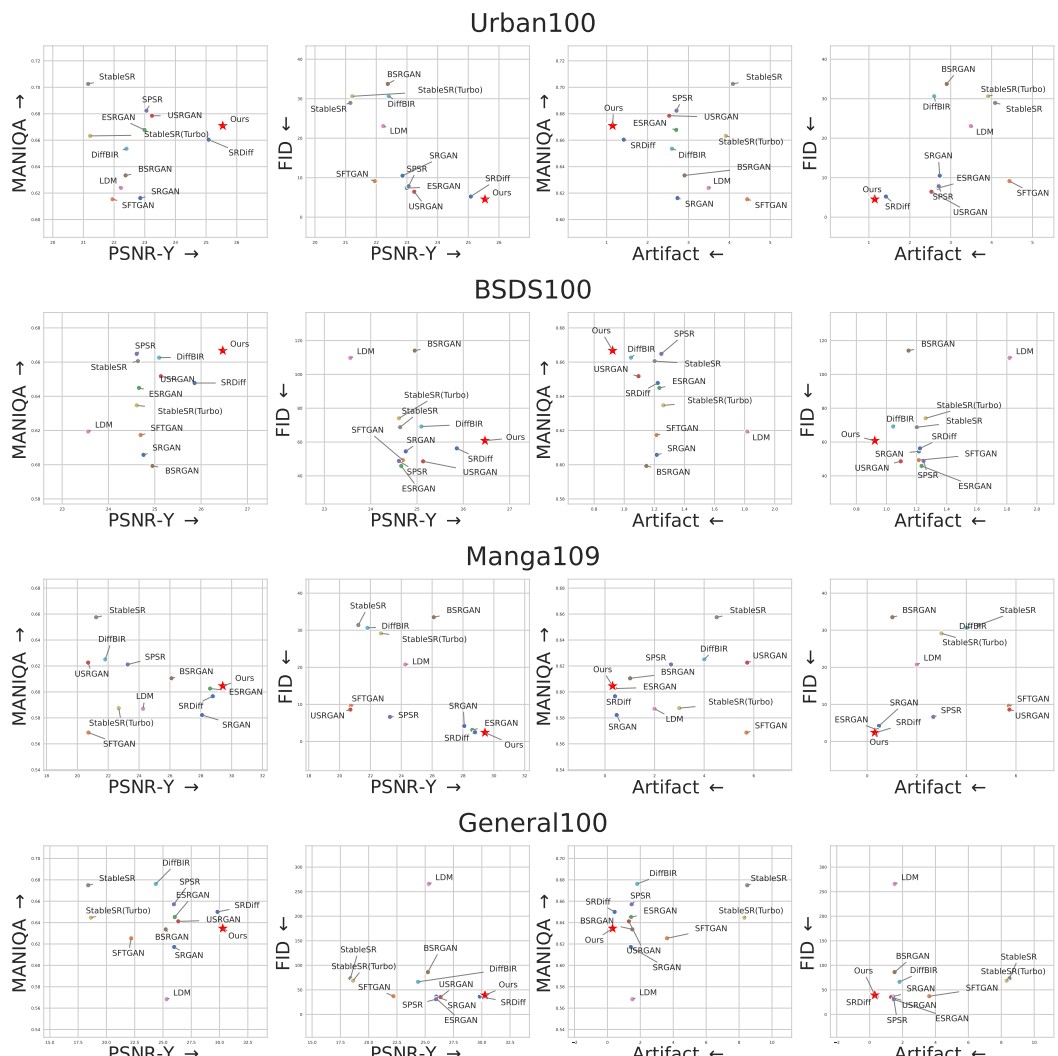

Figure 8: We compared the metrics MANIQA, FID, PSNR, and Artifact across different datasets. In this context, higher values of MANIQA and PSNR are better, while lower values of FID and Artifact are preferred.

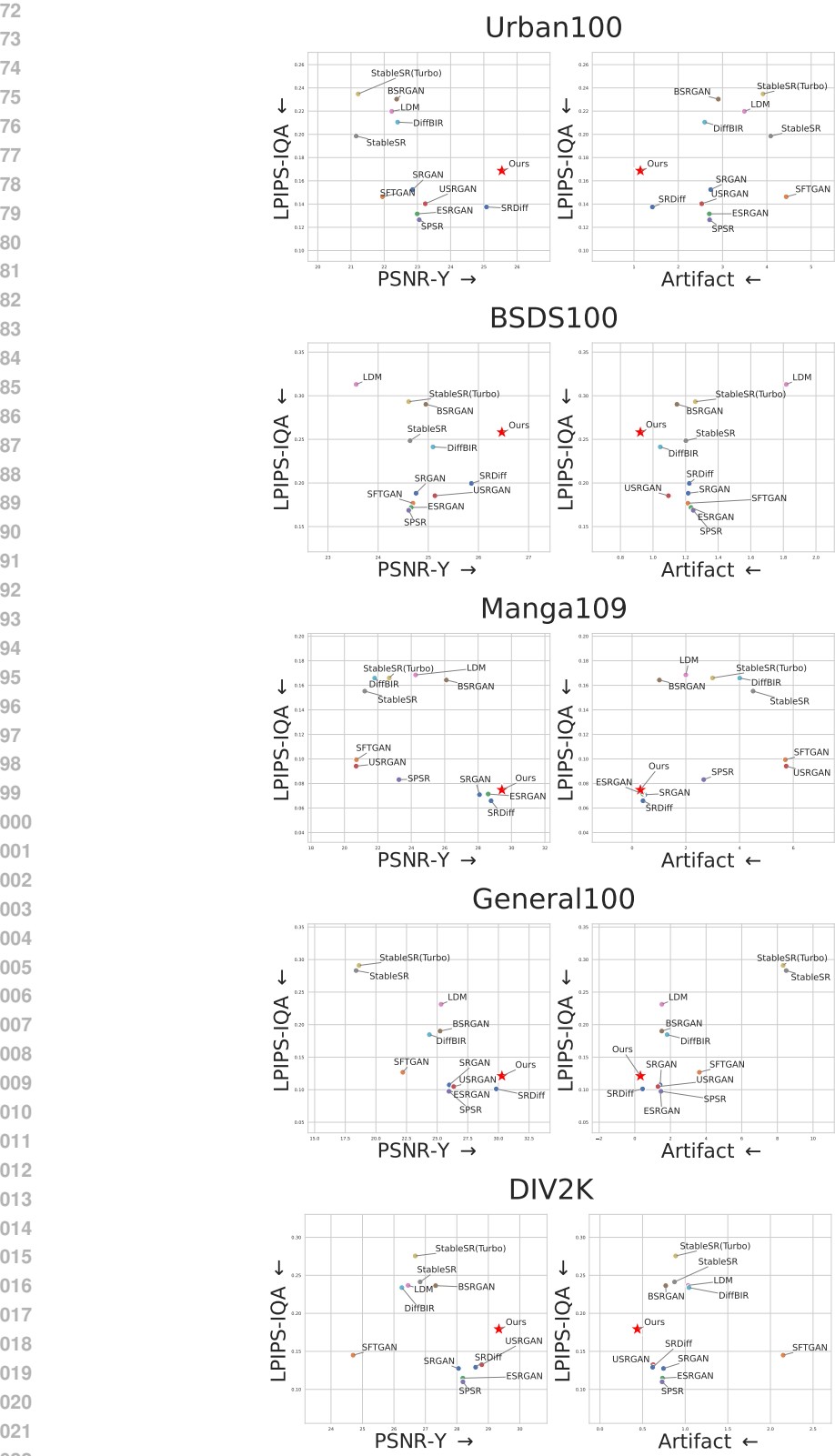

Figure 9: We compared the metrics LPIPS, FID, PSNR, and Artifact across different datasets. In this context, higher values of PSNR is better, while lower values of LPIPS, FID and Artifact are preferred.

