# OpenReview forum: "SAN-Diff: Structure-aware noise for super-resolution diffusion model"
_ICLR.cc/2025/Conference — Submitted to ICLR 2025_

### Official Review · Reviewer_TbBy · 2024-10-20

**Soundness:** 2
**Presentation:** 3
**Contribution:** 2
**Rating:** 5
**Confidence:** 5

**Summary:**

This paper proposes a new model, SAN-Diff, to enhance the performance of image super-resolution (SR) using diffusion models. By leveraging the Segment Anything Model (SAM), SAN-Diff incorporates fine-grained structural information into the noise sampling process, thereby improving the recovery of fine details in images. During training, the model encodes structural position information from SAM into the segmentation mask, which is then used to modulate the sampled noise. The proposed framework integrates this structural information without increasing computational costs during inference.

**Strengths:**

1. The preliminary section is commendably clear and comprehensive, effectively setting the stage for the subsequent content.
2. This paper excels in its theoretical foundation, for providing a meticulous and detailed explanation of the adjusted forward and reverse diffusion processes.
3. The combination of the renowned Segment Anything Model (SAM) and popular diffusion models is particularly interesting.

**Weaknesses:**

1. Almost all of the quotes in the article are in the wrong format. Please refer to the difference between `\citet` and `\citep`.
2. For the super-resolution task, it is suggested to provide general experiments with multiple scales, such as $\times 8 $.
3. Only SRDiff is compared in the Table 1. NOT various diffusion-based image super-resolution methods.

**Questions:**

1. Why not compare the proposed method with common baselines like EDSR and other widely used models?
2. MANIQA, which is a metric for no-reference image quality assessment, seems less common for this super-resolution task. Why not use the more popular LPIPS metric?
3. Since SAM is not used during the inference, so how to utilize the fine-grained structure information from SAM in the inference?
4. To the best of our knowledge, generative models in the super-resolution often obtain lower performance on objective metrics (such as PSNR, SSIM) but higher on subjective ones. Therefore, the higher PSNR is not persuasive.
5. Results in Table 2 (such as SPSR and those of other baselines) are significantly lower than those reported in their original articles. It seems very strange.
6. Some diffusion models like Stable Diffusion (mentioned in the abstract) or ControlNet can also inject structure information (like semantic map) into the diffusion process. Why not compare the results of these models?

---

> ### Author Response · Authors · 2024-11-20
>
> ### **Weaknesses:**
>
> 1. > Almost all of the quotes in the article are in the wrong format. Please refer to the difference between `\citet` and `\citep`.
>
>    Thank you for your suggestion. We have already corrected this error in the revised version.
>
> 2. > For the super-resolution task, it is suggested to provide general experiments with multiple scales, such as ×8.
>
>    Our experimental setup is based on ESRGAN, SRDiff, and DiffBIR, which only tested the X4 setting in their experiments. Therefore, we report the comparison of our method with other models in the X4 setting in Table 2 of the paper. Additionally, in Appendix Table 7, we present the results of SRDiff trained by us and the results of our method in the X2 setting.
>
>    The following are the results for the X8 setting.
>
>    | model           | Urban100 |        |         |       | BSDS100 |        |         |        | Manga109 |        |         |       | General100 |        |         |        | DIV2K  |        |         |        |
>    | --------------- | -------- | ------ | ------- | ----- | ------- | ------ | ------- | ------ | -------- | ------ | ------- | ----- | ---------- | ------ | ------- | ------ | ------ | ------ | ------- | ------ |
>    |                 | PSNR-Y   | SSIM   | MANIQA↑ | FID ↓ | PSNR-Y  | SSIM   | MANIQA↑ | FID ↓  | PSNR-Y   | SSIM   | MANIQA↑ | FID ↓ | PSNR-Y     | SSIM   | MANIQA↑ | FID ↓  | PSNR-Y | SSIM   | MANIQA↑ | FID ↓  |
>    | SRDiff (X8)     | 20.95    | 0.5787 | 0.4375  | 46.71 | 22.74   | 0.5295 | 0.3294  | 155.69 | 23.10    | 0.7471 | 0.3919  | 24.79 | 24.93      | 0.6682 | 0.3756  | 105.39 | 24.54  | 0.6628 | 0.3533  | 8.3945 |
>    | SAN-Diff (X8) | 21.36    | 0.5913 | 0.4188  | 43.75 | 23.29   | 0.5500 | 0.3253  | 154.13 | 23.27    | 0.7520 | 0.3959  | 25.12 | 25.34      | 0.6836 | 0.3648  | 102.17 | 25.18  | 0.6790 | 0.3430  | 8.5189 |
>
> 3. >  Only SRDiff is compared in the Table 1. NOT various diffusion-based image super-resolution methods.
>
>    In Table 2, we report the results of two types of super-resolution models: GAN-based models (SRGAN, SFTGAN, ESRGAN, USRGAN, SPSR, BSRGAN) and diffusion-based models (LDM, StableSR, DiffBIR, SRDiff). We also conduct a fair comparison with our method across multiple metrics. Please refer to Table 2 for the detailed numerical results.

---

> ### Author Response · Authors · 2024-11-20
>
> ### **Questions:**
>
> 1. > Why not compare the proposed method with common baselines like EDSR and other widely used models?
>
>    Thank you for providing this valuable comment. In general, image super-resolution approaches can be categorized into distance-based(like EDSR) and generative-based(like diffusion-based and GAN-base) methods. Since our method focuses on improving diffusion-based SR models, we primarily compare it with generative SR models (GAN-based and diffusion-based) rather than distance-based SR models like EDSR.
>
>    As the official pre-trained weights for EDSR are no longer available, we used the pre-trained weights provided by BasicSR for testing. Below are the comparison results between our model and EDSR.
>
>    | model      | Urban100 |        |         |       | BSDS100 |        |         |       | Manga109 |        |         |       | General100 |        |         |       | DIV2K  |        |         |        |
>    | ---------- | -------- | ------ | ------- | ----- | ------- | ------ | ------- | ----- | -------- | ------ | ------- | ----- | ---------- | ------ | ------- | ----- | ------ | ------ | ------- | ------ |
>    |            | PSNR-Y   | SSIM   | MANIQA↑ | FID ↓ | PSNR-Y  | SSIM   | MANIQA↑ | FID ↓ | PSNR-Y   | SSIM   | MANIQA↑ | FID ↓ | PSNR-Y     | SSIM   | MANIQA↑ | FID ↓ | PSNR-Y | SSIM   | MANIQA↑ | FID ↓  |
>    | EDSR_L     | 24.43    | 0.7500 | 0.4576  | 5.86  | 26.34   | 0.7142 | 0.3231  | 91.85 | 23.99    | 0.8243 | 0.4256  | 5.84  | 27.22      | 0.7971 | 0.4204  | 54.61 | 30.75  | 0.8442 | 0.3555  | 0.3303 |
>    | EDSR_M     | 24.16    | 0.7374 | 0.4297  | 7.23  | 26.29   | 0.7099 | 0.3037  | 96.27 | 24.05    | 0.8214 | 0.4044  | 6.24  | 27.14      | 0.7944 | 0.3986  | 55.39 | 30.44  | 0.8366 | 0.3428  | 0.3456 |
>    | SAN-Diff | 25.54    | 0.7721 | 0.6709  | 4.53  | 26.47   | 0.7003 | 0.6667  | 60.81 | 29.43    | 0.8899 | 0.6046  | 2.40  | 30.29      | 0.8353 | 0.6346  | 39.15 | 29.34  | 0.8108 | 0.5959  | 0.3809 |
>
> 2. > MANIQA, which is a metric for no-reference image quality assessment, seems less common for this super-resolution task. Why not use the more popular LPIPS metric?
>
>    Thank you for providing this valuable comment. We chose to use MANIQA instead of LPIPS to evaluate our method because we believe that MANIQA provides a more accurate assessment of super-resolution image quality. MANIQA evaluates image quality from multiple dimensions [1], including color consistency, structural details, and blurriness. In contrast, LPIPS primarily focuses on perceptual similarity between images. While LPIPS effectively measures differences in image features, it is less capable of analyzing overall image quality. Therefore, to accurately assess the differences in texture and structure between the reconstructed images and HR images, we opted to use MANIQA as a reference metric.
>
> To provide a more comprehensive and objective evaluation of our method, we combined subjective metrics (LPIPS), objective metrics (PSNR), and the number of artifacts to create image, and you can find it in **[LPIPS.png](https://www.helloimg.com/i/2024/11/20/673d8eb72db3f.png)** or in **Figure 9** at appendix, which illustrates a holistic assessment of the model's performance. These results highlight the outstanding performance of our model under combined evaluation criteria.
>
>
>
>    [1] Maniqa: Multi-dimension attention network for no-reference image quality assessment. In *Proceedings of the IEEE/CVF Conference on Computer Vision and Pattern Recognition*.

---

> > ### Author Response · Authors · 2024-11-20
> >
> > 3. > Since SAM is not used during the inference, so how to utilize the fine-grained structure information from SAM in the inference?
> >
> >    Thank you for providing this valuable comment. During training, the U-Net model is trained to predict the added noise modulated by $E_{SAM}$(generate by SAM and SPE module) with the LR image and time step as the condition. During inference, the trained U-Net model can predict the added noise by taking structural information into consideration without requiring $E_{SAM}$
> >
> >    Since the structural knowledge is already injected into the U-Net model during training, the U-Net can leverage the fine-grained structure segmentation ability learned during training to guide image restoration during inference.
> >
> >
> >
> > 4. > To the best of our knowledge, generative models in the super-resolution often obtain lower performance on objective metrics (such as PSNR, SSIM) but higher on subjective ones. Therefore, the higher PSNR is not persuasive.
> >
> >    Our goal is to maintain the advantages of diffusion-based models in subjective metrics while addressing their common issues, such as distorted structures, misaligned textures, and undesirable artifacts. At the same time, we aim to enhance the accuracy of structural and texture reconstruction. This goal is reflected in our method, which achieves a significant improvement in objective metrics (PSNR, SSIM) compared to baselines and other generative models, while maintaining similar performance in subjective metrics.
> >
> >    The combination of subjective metrics, objective metrics, and artifact analysis collectively reflects the accuracy and visual quality of the reconstructed images. To properly evaluate the quality of reconstructed images, it is necessary to assess the model's performance using these three types of metrics comprehensively. Figures 2 and 7 illustrate that our model demonstrates excellent performance under this comprehensive evaluation.
> >
> >
> >
> > 5. > Results in Table 2 (such as SPSR and those of other baselines) are significantly lower than those reported in their original articles. It seems very strange.
> >
> >    We reimplemented SRGAN and ESRGAN using the MMagic framework. For SPSR, SFTGAN, USRGAN, SPSR, BSRGAN, DiffBIR, StableSR, and SRDiff, we reproduced the results using the official code and pretrained weights. For LDM, we performed inference using the publicly available weights on Hugging Face.
> >
> >    The inconsistencies in reported PSNR/SSIM results arise from two primary reasons:
> >
> >    - **Different evaluation codes**: Different models employ different evaluation implementations—some use MATLAB-based code, while others rely on custom Python implementations. These variations introduce discrepancies in the final results.
> >    - **Different testing settings**: Some models compute PSNR and SSIM in the RGB color space, while others convert to the YUV color space and perform evaluations on the Y-channel.
> >
> >    To ensure fair comparisons across models and eliminate the inconsistencies caused by diverse evaluation protocols, we reproduced other models and evaluated their performance under a unified setting using standardized evaluation scripts.
> >
> >    We utilized **IQA-PyTorch**, an open-source image quality assessment tool that provides comprehensive evaluation metrics. All models were assessed using IQA-PyTorch under the same testing settings. This standardization led to differences between the results reported in our paper and those in the original papers.

---

> > ### Author Response · Authors · 2024-11-20
> >
> > 6. > Some diffusion models like Stable Diffusion (mentioned in the abstract) or ControlNet can also inject structure information (like semantic map) into the diffusion process. Why not compare the results of these models?
> >
> >    Thank you for raising this issue. We did not compare our method with Stable Diffusion and ControlNet because these models differ significantly from our experimental settings and objectives, making a direct comparison neither reasonable nor fair.
> >
> >    Since ControlNet is an extension of Stable Diffusion, we use it as a reference to explain the differences between our approaches. ControlNet is a method for injecting conditional information into the model, which incurs additional computational costs during inference. In contrast, our framework does not modify the inference process.
> >
> >    Additionally, we compared our method with StableSR and DiffBIR in the paper. Both methods leverage pre-trained Stable Diffusion as their base model and are specifically fine-tuned for super-resolution tasks. Therefore, we believe that comparing our approach with StableSR and DiffBIR in the context of super-resolution is more reasonable and fair.
> >
> >    ---
> >
> >    We would like to provide a more detailed explanation below.
> >
> >    ControlNet shows that providing a model with additional guiding information can help generate higher-quality images that better meet user expectations. However, a key limitation of ControlNet and similar methods is that they require both the original input (prompt) and auxiliary information (condition) as inputs during both training and inference. For example, in the image SR process using ControlNet, the condition is introduced as follows:
> >
> >    1. An auxiliary model is used to obtain the segmentation mask (condition) of the low-resolution (LR) image.
> >    2. The LR image is then used as input for the diffusion model, with the condition injected into the diffusion process via ControlNet.
> >
> >    This process requires not only ControlNet and the diffusion model but also an additional auxiliary model to generate the condition, which significantly impacts inference efficiency.
> >
> >    Our method addresses this challenge by focusing on the following question: *Is it possible to incorporate the ability to extract condition information directly into the model during training, thereby eliminating the need for an auxiliary model to generate the condition during inference?*
> >
> >    In our paper, we propose to incorporate condition information into the training objective of the diffusion model. As a result, the trained diffusion model can make predictions while considering the condition information, without the need to invoke a segmentation model for test samples during inference. This approach is more efficient than ControlNet.
> >
> >    Regarding the SAM model, it serves as the auxiliary model to obtain the condition information. Intuitively, a more precise segmentation mask provides better guidance for the reconstruction process. We chose SAM over other segmentation models because of its superior performance.

---

> > ### Comment · Reviewer_TbBy · 2024-11-21
> >
> > ## For Question
> > - The values of the metrics reported in this paper appear to **diverge significantly** from the generally accepted results. For instance, recent methods typically report PSNR values (on the Y-channel) of approximately 27-28 for the [Urban100 (x4) dataset](https://paperswithcode.com/sota/image-super-resolution-on-urban100-4x), around 31-33 for the [Manga109 (x4) dataset](https://paperswithcode.com/sota/image-super-resolution-on-manga109-4x), about 28-29 for the [General100 (x4) dataset](https://paperswithcode.com/sota/image-super-resolution-on-general100-4x), and roughly 28 for the [BSD100 (x4) dataset](https://paperswithcode.com/sota/image-super-resolution-on-bsd100-4x-upscaling). Different evaluation codes may lead to small discrepancies, but it cannot account for such so huge deviation (>2 dB for the same model and dataset).
> > - Where is Figure 9 at appendix? This paper only has 7 figures including the appendix.

---

> > > ### Author Response · Authors · 2024-11-21
> > >
> > > Thank you for your response.
> > >
> > > 1. The PSNR scores you referred to are some of the top-ranking results on **Paperswithcode**, which predominantly feature distance-based SR models. As you mentioned:
> > >
> > >    > Generative models in super-resolution often achieve lower performance on objective metrics (such as PSNR and SSIM) but excel in subjective evaluations.
> > >
> > >    The strength of generative models lies in subjective quality rather than objective metrics. Since our method specifically focuses on improving diffusion-based models, directly comparing our results to distance-based SR models on PSNR is neither fair nor meaningful.
> > >
> > >    We used the open-source **IQA-PyTorch** toolkit to evaluate the performance of the relevant models. The scores reported in our paper are reproducible under this unified evaluation setup.
> > >
> > >    It is important to emphasize that our approach aims to improve the structural and textural accuracy of diffusion-based SR models during image reconstruction. This does not imply that our goal is to compete with distance-based SR models for state-of-the-art performance on PSNR or similar metrics.
> > >
> > >    Therefore, we focus on comparing the performance gains of our method against the baseline (SRDiff) and other generative models. To ensure fairness, we conducted comprehensive comparisons with GAN-based and diffusion-based SR models across various metrics, demonstrating the effectiveness of our method.
> > >
> > > 2. We have updated the paper. You can download the revised PDF to find Figure 9 or directly view it here: **[LPIPS.png](https://www.helloimg.com/i/2024/11/20/673d8eb72db3f.png)**.

---

> > > > ### Comment · Reviewer_TbBy · 2024-11-30
> > > >
> > > > I thank the authors for their thoughtful rebuttal. But I am still confused about the authors' viewpoint of objective metrics. At first, they claim that their method `achieves a significant improvement in objective metrics (PSNR, SSIM) compared to baselines and other generative models` (Note that the metrics of these baselines reported in this paper appear to diverge significantly from the generally accepted results). This point is regarded as one important contribution in this paper (see Abstract and Introduction). Then, they claim that comparing on PSNR is neither fair nor meaningful. I am afraid it may seem contradictory.

---

> ### Comment · Reviewer_TbBy · 2024-11-21
>
> ## For Weakness:
> - What is the meaning of SAM-DiffSR in this comment? Do you mean SAN-Diff?
> - For Table 1, could you give more comparisons of the effectiveness and efficiency about other **recent** diffusion-based methods? Note that SRDiff was published in 2022, which is not so new.

---

> > ### Author Response · Authors · 2024-11-21
> >
> > Thank you for your response.
> >
> > 1. We sincerely apologize for the confusion caused by our writing error. The correct name should be SAN-DiFF. Thank you for pointing this out. We have corrected the method name in the table.
> >
> > 2. It is worth noting that the original Table 1 is an ablation study where we report three sets of results: SRDiff, SRDiff with SAM directly integrated during training and inference, and our method. Since our approach is an improvement based on SRDiff, this experiment aims to demonstrate two key points:
> >
> >    - The comparison between SRDiff and SAM+SRDiff shows that incorporating additional segmentation information into diffusion models can improve performance.
> >    - The comparison among SRDiff, SAM+SRDiff, and SAN-DiFF demonstrates that our method achieves performance improvements similar to SAM+SRDiff, without increasing training or inference time.
> >
> >    Thus, Table 1 focuses on comparisons with SRDiff and SAM+SRDiff.
> >
> >    To provide more comprehensive insights, we also include comparisons with other diffusion models. However, since training times are recorded on different hardware setups, they are provided for reference only. All other evaluations were conducted using a V100 GPU on the DIV2K dataset.
> >
> >    |                | SRDiff         | SAM+SRDiff     | SAN-Diff       | LDM           | StableSR                              | DiffBIR                              |
> >    | -------------- | -------------- | -------------- | -------------- | ------------- | ------------------------------------- | ------------------------------------ |
> >    | Parameter      | 12M            | 644M           | 12M            | 169M          | 960M                                  | 1101.8M                              |
> >    | Train time     | 2 days         | 10 days        | 2 days         | 6 days        | 10 days + Stable Diffusion train time | 6 days + Stable Diffusion train time |
> >    | Inference time | 37.64s/per img | 65.72s/per img | 37.62s/per img | 26.3s/per img | 238.6s/per img                        | 112.4s/per img                       |
> >    | PSNR           | 28.6           | 29.41          | 29.34          | 26.45         | 26.83                                 | 26.25                                |
> >    | FID            | 0.4649         | 0.3938         | 0.3809         | 9.5518        | 14.5232                               | 17.8206                              |

---

> ### Author Response · Authors · 2024-11-25
>
> Dear TbBy,
>
> Thank you for your constructive comments and valuable suggestions to improve this paper. ​If you have any more questions, we would be glad to discuss them with you.
>
> Thank you very much.
>
> Best regards, Author

---

> ### Author Response · Authors · 2024-11-27
>
> Dear TbBy,
>
> We apologize for the repeated reminder.  It has been 6 days since we submitted our responses, but we have not yet received your feedback.  **We simply want to ensure that we have fully addressed all your concerns.**
>
> Your main concerns appear to relate to the values of the metrics reported and the comparisons of the effectiveness and efficiency about other recent diffusion-based methods.  **We have provided detailed responses to these questions in both our reply to you and the general response.**  May we kindly ask if you could spare some time to review our responses and share your feedback?
>
> If there are any remaining points that require further clarification, please rest assured that we are committed to providing detailed answers to all your inquiries and addressing any concerns you may have. We value clear and open communication, and will make every effort to ensure that all aspects of the matter are fully explained to your satisfaction.
>
> Thank you very much.
>
> Best regards, Author

---

> ### Author Response · Authors · 2024-11-30
>
> Thank you for your reply and constructive questions. There are a few foundational concepts that need clarification.
>
> Current super-resolution (SR) models can be classified into two categories:
>
> - **Distance-based SR models:** Examples include EDSR[3], SwinIR[4], HAT[5], and DRCT[1]. These typically use distance loss (e.g. $L_1$ pixel loss) to train the model.
> - **Generate-based SR models:**
>   - **GAN-based models**: Examples like ESRGAN[6], which usually combine distance loss with additional perception loss, and are trained using a generative adversarial approach.
>   - **Diffusion-based models**: Examples include SRDiff[7], LDM[8], and StableSR[2], which generally use MSE loss and are trained through the denoising process.
>
> The difference in training methods results in distinct advantages for distance-based and generate-based models in terms of image reconstruction. **Distance-based models**, due to the constraint of distance loss, **tend to reconstruct images with more accurate structures but less defined textures**. On the other hand, **generate-based models**, due to their different training processes and losses, **tend to produce images with clearer textures but may exhibit distorted structures**. Therefore, in the field of single image super-resolution (SISR), these two categories are typically compared separately.
>
> For example, taking **DRCT [1]**, the highest-scoring model on the [Urban100](https://paperswithcode.com/sota/image-super-resolution-on-urban100-4x) benchmark, it primarily compares models like EDSR[3], SwinIR[4], and HAT[5] in their paper, which are all **distance-based SR models**. In contrast, **StableSR[2]** compares models like BSRGAN[9] and LDM[8] in paper, which are **generate-based SR models**.
>
> We would also like to reiterate that comparing our method with distance-based SR models on their stronger metrics (PSNR and SSIM) is not a fair comparison. **The conventional and fair approach is to compare our method with generate-based SR models across all relevant metrics**. Our goal is to improve the PSNR and SSIM scores of diffusion-based SR models, thereby enhancing the accuracy of structure and texture reconstruction. Our experimental results validate the effectiveness of our approach.
>
> ---
>
> Regarding the discrepancy in metrics compared to the original papers, we would like to reiterate that **all the data reported in our paper** (whether for our method or others) **can be consistently reproduced using open-source tools.**
>
> The field of super-resolution has a long development history, and over time, there have been numerous changes in the evaluation settings and scripts used to assess model performance. This makes it difficult to directly compare models using the data reported in original papers in an intuitive and fair manner. Therefore, we have chosen to use a unified open-source evaluation tool to fairly assess all methods. This allows readers to make an intuitive and accurate comparison between our method and the others without needing to switch between different testing settings.
>
> ---
>
> [1] DRCT: Saving Image Super-resolution away from Information Bottleneck[J]. arXiv preprint arXiv:2404.00722, 2024.
>
> [2] Exploiting diffusion prior for real-world image super-resolution[J]. International Journal of Computer Vision, 2024: 1-21.
>
> [3] Enhanced deep residual networks for single image super-resolution[C]//Proceedings of the IEEE conference on computer vision and pattern recognition workshops. 2017: 136-144.
>
> [4] Swinir: Image restoration using swin transformer[C]//Proceedings of the IEEE/CVF international conference on computer vision. 2021: 1833-1844.
>
> [5] Activating more pixels in image super-resolution transformer[C]//Proceedings of the IEEE/CVF conference on computer vision and pattern recognition. 2023: 22367-22377.
>
> [6] Enhanced super-resolution generative adversarial networks[C]//Proceedings of the European conference on computer vision (ECCV) workshops. 2018: 0-0.
>
> [7] Srdiff: Single image super-resolution with diffusion probabilistic models[J]. Neurocomputing, 2022, 479: 47-59.
>
> [8] High-resolution image synthesis with latent diffusion models[C]//Proceedings of the IEEE/CVF conference on computer vision and pattern recognition. 2022: 10684-10695.
>
> [9] Designing a practical degradation model for deep blind image super-resolution[C]//Proceedings of the IEEE/CVF International Conference on Computer Vision. 2021: 4791-4800.

---

### Official Review · Reviewer_aaKr · 2024-11-01

**Soundness:** 3
**Presentation:** 2
**Contribution:** 3
**Rating:** 6
**Confidence:** 3

**Summary:**

The authors proposed SAN-Diff for image super-resolution task. The authors noticed that one potential limitation of existing works could be sampling from one distribution during the diffusion process, and might hurt the generated image results when the scene is complex. They proposed to using a positional embedding, calculated with respect to regions segmented by SAM, as the condition of an existing work Srdiff, and achieved good performance, on multiple datasets.

**Strengths:**

The authors’ proposal sound valid as the spatially uniform sampling could be a place where the potential improvement can happen. The math deduction looks valid. The computation cost caused by the proposed design seems neglectable, in comparison with the original approach. Experimentations of different model comparisons and ablation studies are comprehensive.

**Weaknesses:**

It seems like the authors have a strong assumption that the SAM always provides accurate segmentation results. Although it is a powerful segmentation model could be used to zero-shot inference on everyday object, it is “statistically” strong. Therefore, your experimentations also show quantitatively “statistically” better. It would be great if the authors could discuss about the limitations of using SAM as the segmentation mask provider.

**Questions:**

1. For the example shown in Fig 5,6, is it possible for the authors provide the segmentation masks using SAM? Then, it will help us to see if the segmentation mask provided by SAM matches the “better” parts of the SR images you generated, and justify your conclusions.
2. In Figure 1, it is hard for me to think of why this improvement happens because of the contribution from SAM. I highly doubt that the image region you contoured can be identified as different “regions” segmented by SAM. How would the authors justify the contribution of SAM?
3. In Table 2, I see the proposed model performed the best. However, the FID score for SRDiff and SAN-Diff suddenly dropped a lot across different datasets (the last two rows), which causes my concern that if the authors didn’t choose good enough baselines. Please justify. I also would like to refer other reviewers’ comments.

---

> ### Author Response · Authors · 2024-11-20
>
> ### **Weaknesses:**
> > the limitations of relying on SAM as the sole provider of segmentation masks
>
> Thank you for your valuable comments. Compared to the original diffusion model without structural guidance, masks generated by existing SAM models can improve performance, as demonstrated in our experimental results.
>
> However, the performance of our model does depend on the quality of the segmentation masks, as they capture the structural information of the corresponding image. Our model benefits from SAM's fine-grained segmentation capability and its strong generalization ability across diverse objects and textures in the real world. Nevertheless, the performance of our model is also limited by the capabilities of the segmentation model itself. For instance, SAM may struggle to identify structures with low resolution in certain scenes. While the model can partially mitigate this issue by learning from a large amount of data during training, it is undeniable that higher segmentation precision (e.g., SAM2) and finer segmentation granularity would significantly enhance the performance of our approach. This is one of the potential limitations of our proposed framework, and we have included this discussion in the revised version of our paper in Appendix C.3.
>
> Our experiments demonstrate that even though existing segmentation models may not perfectly distinguish every region in the real world, combining our method with a super-resolution model has substantially improved performance on super-resolution tasks. We believe that utilizing prior knowledge from segmentation models to enhance generative models is a promising avenue for further exploration.

---

> ### Author Response · Authors · 2024-11-20
>
> ### **Questions:**
>
> 1. > For the example shown in Fig 5,6, is it possible for the authors provide the segmentation masks using SAM?
>
>    Figure 7 shows the segmentation results from SAM. However, it is important to emphasize that our method only uses the mask information during the training process, and no mask guidance is used during inference. It can serve as a reference for analyzing the super-resolution results.
>
> 2. > In Figure 1, it is hard for me to think of why this improvement happens because of the contribution from SAM. I highly doubt that the image region you contoured can be identified as different “regions” segmented by SAM. How would the authors justify the contribution of SAM?
>
>    We aim to augment diffusion-based image super-resolution (SR) models with fine-grained masks provided by SAM. In our paper, we compare two designs. The first involves using the information provided by SAM as a conditioning factor for the U-Net, following a commonly used approach in guided diffusion. The second design, structural noise modulation, is our heuristic approach, with the derivation provided in the supplementary materials. From the results in Figure 1(B), both qualitative and quantitative results demonstrate the effectiveness of incorporating segmentation information to enhance the model’s capabilities, as well as the fact that structural noise modulation yields better performance without adding extra inference cost.
>
>    Furthermore, as you mentioned, the model learns information "statistically."   Although there may be some regions where the segmentation model's performance is limited and it might fail to distinguish or incorrectly segment certain areas, the model learns from a large amount of data during training.   The denoising model (U-Net) can "statistically" learn the necessary structure-level ability to distinguish different regions.   Therefore, these issues do not affect the model to get ability to distinguish regions during training, and they do not prevent the model from reconstructing more accurate structures and textures during inference.
>
>    A similar example can be seen in classification tasks using the ImageNet dataset. According to [1], ImageNet contains a significant number of mislabeled samples, samples with multiple labels, or samples that do not belong to any label. This is analogous to the regions SAM fails to correctly identify. These mislabeled data points are essentially "negative samples" in the training dataset. However, this does not hinder the training of classification models like ResNet or ViT, as the model learns "statistically" during training. It can self-correct learned errors in labels and predict the correct results during inference.
>
>    [1] Northcutt, C. G., Athalye, A., & Mueller, J. (2021). Pervasive label errors in test sets destabilize machine learning benchmarks. *arXiv preprint arXiv:2103.14749*.
>
> 3. > In Table 2, I see the proposed model performed the best. However, the FID score for SRDiff and SAN-Diff suddenly dropped a lot across different datasets (the last two rows), which causes my concern that if the authors didn’t choose good enough baselines. Please justify. I also would like to refer other reviewers’ comments.
>
>    Our goal is to maintain the advantages of diffusion-based models in subjective metrics while reducing common issues such as distorted structures, misaligned textures, and bothersome artifacts. We aim to improve the accuracy of structural and texture reconstruction in the model. This goal is reflected in our results, where we significantly improve objective metrics (PSNR, SSIM) while maintaining similar subjective metric performance (as shown in the last two rows) compared to the baseline and other generative models. Additionally, subjective metrics, objective metrics, and the number of artifacts all reflect the accuracy and visual quality of the reconstructed images. To properly assess the quality of the reconstructed images, it is necessary to evaluate the model's performance based on all three types of metrics. In Figure 2 and Figure 8, we present images that highlight the excellent performance of our model based on this comprehensive evaluation.

---

> > ### Comment · Reviewer_aaKr · 2024-11-22
> >
> > Thanks for the reply. I would like to leave my score unchanged.

---

### Official Review · Reviewer_mefZ · 2024-11-02

**Soundness:** 3
**Presentation:** 3
**Contribution:** 3
**Rating:** 6
**Confidence:** 5

**Summary:**

This paper proposes SAN-Diff, a structure-aware noise modulation method for super-resolution (SR) diffusion models. SAN-Diff uses the Segment Anything Model (SAM) to generate fine-grained segmentation masks, which guide the noise modulation process, allowing the diffusion model to apply distinct noise distributions to different areas of the image. The Structural Position Encoding (SPE) module is introduced to integrate position information into these masks.

**Strengths:**

1. By modulating noise based on segmentation areas, SAN-Diff enhances the preservation of local structures, particularly useful for complex scenes and intricate textures.
2. SAN-Diff uses SAM-generated masks only during training, avoiding computational burdens during inference while retaining structure-level detail restoration.

**Weaknesses:**

1.  Lack of Novelty in Methodology: The SAN-Diff approach heavily relies on existing methods, including the Segment Anything Model (SAM) for segmentation and structural positional encoding (SPE) techniques. This reliance limits the novelty of the proposed methodology, without introducing fundamentally new innovations tailored for super-resolution tasks.
2. Limited Novelty in Designing Loss Function: The loss function shown in Eq.5 mainly follows a standard existing MSE approach focused on noise prediction. Only the structurally positioned embedded mask is just added  to it.
3. Limited Exploration of Alternative Modulation Strategies: SAN-Diff exclusively uses a segmentation-mask-driven approach for noise modulation without investigating other methods, such as feature-based techniques.
4. No Discussion on the Time Complexity of SPE: The paper mentions negligible cost but does not clarify the computational time taken by  SPE module for generating masks before training.
5. Dependency on High-Resolution Training Data: The method requires SAM-generated segmentation masks for all high-resolution (HR) training images, which may not always be feasible in datasets lacking extensive HR samples or where SAM’s segmentation quality is inconsistent. This reliance could restrict SAN-Diff’s application in low-data or low-resolution scenarios.

**Questions:**

- Could the authors mention if there are any unique aspects in how SAM and SPE are integrated or applied that are tailored for this task?
- Could the authors try using the other losses like structural loss instead of standard MSE loss for better results?
- Could the authors mention if they tried alternative feature extraction techniques like -depth maps, CNN features etc. instead of segmentation masks.
- Could the authors provide concrete timing measurements or complexity analysis for the SPE module. This would help to understand the practical implications of using this approach.

---

> ### Author Response · Authors · 2024-11-20
>
> ### **Weaknesses:**
> 1. > Lack of Novelty in Methodology
>
>    We sincerely appreciate your thoughtful comment. We would like to clarify that the **Structural Position Encoding (SPE)** module is a novel approach designed to encode structural position information within the masks generated by segmentation models. This module is introduced for the first time in our paper and **is not based on any pre-existing method**. Regarding the SAM model, we use it to extract structural information for structural noise modulation within our framework. This design is flexible and does not restrict the source of segmentation masks, meaning that segmentation models other than SAM can also be utilized. We have chosen SAM specifically due to its exceptional ability to generate fine-grained segmentation masks.
>
> 2. > Limited Novelty in Designing Loss Function
>
>    The key modification to the loss function lies not in the MSE metric itself, but in the object to which the distance is measured. In our proposed framework, we closely follow the DDPM setting [1] within the context of the noise modulation scenario, leading to the formulation of the modified loss function. By training with this modified loss, the denoising model is able to incorporate structural information, resulting in improved SR image generation. Our experimental results demonstrate the effectiveness of this design.
>
>    [1] Denoising diffusion probabilistic models. *Advances in neural information processing systems*, *33*, 6840-6851.
>
> 3. > Limited Exploration of Alternative Modulation Strategies
>
>    The primary aim of our work is to enable the model to achieve structure-level differentiation between regions. To this end, we use segmentation masks as the source of structural information for training, as they inherently contain rich structural details. While intermediate features may also carry abundant information, they are less directly related to image structure and more challenging to utilize effectively. Therefore, we have focused on a segmentation-mask-driven approach in this study. In future work, we plan to explore feature-based techniques to further enhance model performance.
>
> 4. > No Discussion on the Time Complexity of SPE
>
>    Generating segmentation masks on the DIV2K dataset using SAM takes approximately 4 hours on GPU, while synthesizing SPE masks with the SPE module requires only 15 minutes on CPU. These data do not need to be regenerated for every training session; instead, a single inference of the training dataset using the segmentation model can be performed beforehand, and the results can be reused across different training runs. Therefore, we consider the cost of this one-time preprocessing negligible compared to the total training time of approximately 50 hours.
>
> 5. > Dependency on High-Resolution Training Data
>
>    In the current literature on image super-resolution, the standard approach involves training super-resolution models using pairs of low- and high-resolution data. However, there has been little to no exploration of high-resolution image-free super-resolution tasks. We believe this is an interesting task and leave it for future research.

---

> ### Author Response · Authors · 2024-11-20
>
> ### **Questions:**
>
> 1. > Could the authors mention if there are any unique aspects in how SAM and SPE are integrated or applied that are tailored for this task?
>
>    We leverage the powerful fine-grained segmentation capabilities of SAM to modulate the noise distribution across different regions. The fundamental concept behind the SPE module is to assign a unique value to each segmentation area. The segmentation mask generated by SAM comprises a series of 0-1 masks, where each mask corresponds to an area in the original image sharing the same semantic information.  To achieve noise modulation based on mask information, we merge these masks using the SPE module. Inspired by successful practices in language models, we use RoPE to encode the masks at different positions, embedding the relative positional information of different segmentation areas into the noise.
>
> 2. > Could the authors try using the other losses like structural loss instead of standard MSE loss for better results?
>
>    Our modification to the loss function lies not in the MSE metric itself, but in the object to which the distance is measured. While adopting other metrics, such as MAE, may also be effective, this is beyond the scope of our paper’s focus. Furthermore, to ensure a fair comparison with existing works [1] that consistently use the MSE loss, we have adhered to the same configuration in our experiments.
>
>    [1] Denoising diffusion probabilistic models. *Advances in neural information processing systems*, *33*, 6840-6851.
>
> 3. > Could the authors mention if they tried alternative feature extraction techniques like -depth maps, CNN features etc. instead of segmentation masks.
>
>    Our goal is to enhance the accuracy of structural and texture reconstruction in diffusion models without compromising their advantages in subjective metrics. The most intuitive approach is to leverage the structure-level ability of segmentation models to distinguish different regions. We believe that combining depth maps or CNN features with our method could further improve the model's performance, and we look forward to related research in the future.
>
> 4. > Could the authors provide concrete timing measurements or complexity analysis for the SPE module. This would help to understand the practical implications of using this approach.
>
>    Synthesizing SPE masks on the DIV2K dataset using the SPE module takes only 15 minutes. Moreover, these masks do not need to be regenerated for each training session. A single inference using the segmentation model on the training dataset, followed by saving the results, allows these masks to be reused across different training runs.

---

> ### Author Response · Authors · 2024-11-25
>
> Dear mefZ,
>
> Thank you for your constructive comments and valuable suggestions to improve this paper. ​If you have any more questions, we would be glad to discuss them with you.
>
> Thank you very much.
>
> Best regards, Author

---

> ### Author Response · Authors · 2024-11-27
>
> Dear mefZ,
>
> We apologize for the repeated reminder.  It has been 7 days since we submitted our responses, but we have not yet received your feedback.  **We simply want to ensure that we have fully addressed all your concerns.**
>
> Your main concerns appear to relate to our contributions and the dependence on data and computational complexity.  **We have provided detailed responses to these questions in both our reply to you and the general response.**  May we kindly ask if you could spare some time to review our responses and share your feedback?
>
> If there are any remaining points that require further clarification, please rest assured that we are committed to providing detailed answers to all your inquiries and addressing any concerns you may have. We value clear and open communication, and will make every effort to ensure that all aspects of the matter are fully explained to your satisfaction.
>
> Thank you very much.
>
> Best regards, Author

---

### Meta-Review · Area_Chair_FGB5 · 2024-12-17

**Metareview:**

This paper proposes SAN-Diff, a structure-aware noise modulation method for super-resolution (SR) diffusion models.  SAN-Diff uses the Segment Anything Model (SAM) to generate fine-grained segmentation masks, which guide the noise modulation process, allowing the diffusion model to apply distinct noise distributions to different areas of the image.  Additionally, the paper introduces the Structural Position Encoding (SPE) module to integrate position information into these masks.

SAN-Diff enhances the preservation of local structures, particularly useful for complex scenes and intricate textures, by modulating noise based on segmentation areas.  SAN-Diff uses SAM-generated masks only during training, avoiding computational burdens during inference while retaining structure-level detail restoration.  The paper provides a detailed explanation of the adjusted forward and reverse diffusion processes.  The combination of the Segment Anything Model (SAM) and popular diffusion models is interesting.

However, the paper's evaluation presents several limitations:

- Over-reliance on SAM: The method assumes consistently accurate segmentation from SAM, neglecting potential inaccuracies. While the revised version partially addresses this in Appendix C.3, a more central discussion of SAM's limitations and their impact on SAN-Diff is crucial. Visualizing SAM-generated masks for specific examples would further strengthen this analysis.

- Inconsistent Baseline Comparisons: The reported performance of baseline methods appears significantly lower than in their original publications, raising concerns about the consistency of metric computation. Additionally, the chosen baselines seem relatively outdated, and the use of the less common MANIQA metric instead of the widely adopted LPIPS metric further hinders meaningful comparisons.

- Unclear Benefits: While the proposed idea is interesting, the evaluation falls short of convincingly demonstrating the true advantages of SAN-Diff. The limitations mentioned above make it difficult to accurately assess its performance against recent approaches in the field.

In conclusion, SAN-Diff presents a promising direction for structure-aware super-resolution. However, addressing the outlined concerns regarding SAM's reliability, baseline comparisons, and a more robust evaluation would significantly enhance the paper's contribution and clarity.

**Additional Comments On Reviewer Discussion:**

During the rebuttal period, the reviewers raised several concerns, including the novelty of the proposed methodology, the reliance on SAM, the result inconsistencies (metrics), and the lack of comparisons with common baselines. The authors responded to these concerns by clarifying the novelty of the SPE module, explaining the reasons for using SAM, and providing additional experimental results. However, some reviewers remained unconvinced, particularly regarding the novelty and result inconsistencies. In my final decision, I have taken into account the reviewers' concerns and the authors' responses. I have also considered the paper's strengths and weaknesses, as well as its missing components.

---

### Decision · Program_Chairs · 2025-01-22

Reject